

# Model evaluation of high-resolution urban climate simulations: using WRF ARW/LSM/SLUCM model as a case study

Zhiqiang Li[1], Yulun Zhou[2], Bingcheng Wan[3], Bo Huang[1] , Biao Liu[4], and Hopun Chung[5]

[1]Institute of Space and Earth Information Science, The Chinese University of Hong Kong, Hong Kong, 999077, China
[2]Department of Geography and Resource Management, The Chinese University of Hong Kong, Hong Kong, 999077, China
[3]Glarun Technology Co., Ltd., Nanjing, 211100, China
[4]Research Institute of Tsinghua University in Shenzhen, Shenzhen, 518057, China
[5]Department of Computer Science and Engineering, The Chinese University of Hong Kong, Hong Kong, 999077, China

*Correspondence to*: Bo Huang (bohuang@cuhk.edu.hk)

Zhiqiang Li and Yulun Zhou contributed equally to this work and should be considered co-first authors.

**Abstract.** The veracity of urban climate simulation models should be systematically evaluated to demonstrate the trustworthiness of these models against possible model uncertainties. However, existing studies paid insufficient attention to the model evaluation; most studies only provided some simple comparison lines between modelled variables and their corresponding observed ones on the temporal dimension. Challenges remain since such simple comparisons cannot concretely prove that the simulation of urban climate behaviors is reliable. Studies without systematic model evaluations are ambiguous or arbitrary to some extent, which may still lead to some seemingly new findings, but these findings may be scientifically misleading. To tackle these challenges, this article proposes a methodological framework for the model evaluation of high-resolution urban climate simulations and demonstrates its effectiveness with a case study in the fast-urbanizing city of Shenzhen, China. It is intended to remind (again) urban climate modelers of the necessity of conducting systematic model evaluations in urban-scale climatology modelling and reduce these ambiguous or arbitrary modelling practices.

## 1 Introduction

Recently, studies on urban climate have received growing attention, since 66% of the world's population will be living in cities by 2050 (United Nations, 2014) and their fundamental well-being, such as their comfort and health, is directly and significantly affected by urban meteorological conditions, including but not limited to temperature, wind speed and air pollution. Meanwhile, the ongoing global trend of climate change adds to the urgency and significance of achieving better understandings of urban climate and obtaining more precise predictions of future changes and their related impacts. In this vein, many tools have been developed and the rapidly developing urban climate simulation models are among the most powerful ones. These simulation models have been widely applied in analyses and predictions of urban climate conditions, as well as assessments of urban climate impacts brought by the dramatic human interferences in cities.

Model evaluation is necessary for urban climate simulations to make sure the results are reliable and trustworthy to some extent. Urban climate simulation is employed to obtain fine-scale details from the lateral boundary condition of coarse-scale meteorological data by using a limited area model. Moreover, in order to construct precisely the fine-scale details at utmost in the area of interest, the model takes the area of interest land surface forcing into account (Lo et al., 2008). The fine-scale details are constructed by a limited area model consisting of physical components driven by the lateral boundary conditions of coarse-scale meteorological data and land surface forcing data but actually don't exist in the coarse-scale meteorological




data. In its essence, the fine-scale details constructed by a limited area model have the possibility of deviating from their corresponding natural values. Urban climate simulation, with a higher requirement on its resolution (spatial and temporal) and modelling urban climatological phenomena (for example, urban heat island, temperature difference between urban and non-urban areas), is more sensitive to the inadequacies of the atmospheric model, the inappropriate configuration of the

modelling system (Warner, 2011) and the quality of input data. Therefore, model evaluation is even more critical for urban climate simulation.

However, recent efforts understandably paid minimum attention to model evaluation in the community of urban climate modelers, which weakens the reliability of every conclusion based on the insufficiently justified model results. Among existing literature, researchers mostly conducted some simple comparisons between modelled variables and their

corresponding observed ones by drawing their short-term time-history plots. For example, Jiang et al. (2008) made a bold prediction that the near-surface temperature in the Huston area will increase by 2°C in future years (2051–2053). However, the conclusion was only supported by a simple comparison between the observed and WRF-modelled diurnal 2-meters air temperature during August 2001–2003. Meng et al. (2011) modelled the 2-meters air temperature and heat island intensity by using three different modelling schemes and concluded which one is best in modelling performances. However, these

seemingly robust conclusions are only based on a comparison of the observed temperatures with their corresponding modelled ones over a period of 3 days. With a simple model evaluation comparing diurnal patterns of 3-months-WRF-modelled 2-m surface temperature, special humidity, and relative humidity with its corresponding observed ones, Yang et al. (2012) asserted that the WRF model could reconstruct the urban climate features at high resolution of 1-km and had a good performance in modelled surface air temperature and relative humidity in the Nanjing area. Although the afore-mentioned

efforts partially addressed the evaluation issue, significant challenges remain in establishing the trustworthiness of the model: Even if an exact match between a modelled variable at some grids and its corresponding observed one in a period cannot conclude that the model simulates urban climate successfully, not to mention a non-exact match. These model evaluation methods are not convincing, or even reckless. That kind of modelling practices without a convincing model evaluation is still prevalent in climate modelling community even for the most recent literature, such as the papers of Gu and

Yim (2016), Wang et al. (2016) and Bhati and Mohan (2016). Thus, in this paper, we dig deeply into the model evaluation and propose a systematic framework and methods for evaluating model results from multiple perspectives, to benefit future studies with more choices for model quality control and make urban scale simulation more robust. Moreover, we also provide a case analysis of the interval between the modelled atmospheric variable and its corresponding observed one.

The remainder of this paper is organized as follows. Section 2 introduces the proposed framework for model evaluation,

experimental design, and data used for modelling and model evaluation. Section 3 introduces the technical preparation for the urban climate simulation. Section 4 presents various results of the proposed model evaluation methods in our case study. Section 5 concludes the paper with discussions.

## 2 Methodology

### 2.1 Modelling Method

In an urban area, the natural texture of the land surface has remarkably changed to human-made, impervious land surface. The textural change of the land surface leads to modifications in the interchange of energy, momentum, and mass between the land surface and planetary boundary layer (PBL) (Wang et al., 2009). Moreover, in an urban area, the anthropogenic heat release caused by human activities increases the sensible and latent heat emission. Furthermore, the urban building morphology also has an impact on the radiation exchange and the airflow. Tewari et al. (2007) developed the Urban Canopy





Model (UCM) to couple with the Advanced Research WRF (ARW) model via Noah Land Surface Model (Noah-LSM) to improve the simulation accuracy of urban processes by integrating these physical characters below the urban canopy.

We take Shenzhen and Hong Kong, a region in China that had gone through intensive urbanization process, as the study area. WRF ARW model coupled with LSM/SLUCM was used for modelling urban climate in 2010 at 1-km grid spacing.

Through comparison, we found that some of the terrestrial input data provided by NCAR were out-of-date, especially for data describing the fast-developing area. To reflect more precisely the artificial changes on the physical environment brought by the urbanization, we developed four sets of high-resolution urban data, including the vegetation coverage, building morphology, land cover, and anthropogenic heat, and used them as inputs for the follow-up urban climate simulation, and so the simulated urbanization impacts on urban climate would be more accurate.

**2.2 Experimental Design**

Firstly, running an atmospheric model consumes a considerable amount of computational resources, especially for simulating long-term climate. Due to limitations in computational resources, we divided the urban climate simulation case into sequenced four days simulation segments. For each segment, the first day overlaps with the last day of its previous simulation segment, which was used for model spin-up.

**2.3 Data for Model Evaluation**

In existing literature, Numerical Weather Prediction (NWP) models are typically evaluated by comparing the spatial-temporal patterns of the modelled variables with those of its corresponding near-surface observations. Table 1 shows the observation datasets we used for the comparison between modelled results and observations in the inner-most domain. Table 2 lists the modelled variables we included in the model evaluation.

**Table 1: Observation Datasets.**

| Observation Datasets | Sources |
|---|---|
| 2010 PRD Observation Locations | Meteorological Bureau of Shenzhen Municipality |
| 2010 PRD 2-Meters Air Temperature | |
| 2010 PRD 10-Meters Wind Speed | |
| 2010 PRD Precipitation | |
| 2010 PRD Relative Humidity | |
| 2010 MODIS/Aqua Land Surface Temperature and Emissivity (LST/E) product | NASA EOSDIS Land Processes DAAC, USGS Earth Resources Observation and Science (EROS) Center |

**Table 2: Modelled Variables for Model Evaluation.**

| Modelled Variables for Model Evaluation | | Corresponding Observation Datasets |
|---|---|---|
| **Name** | **Description** | |
| T2 | 2-meters air temperature | 2010 PRD 2-Meters Air Temperature |
| TSK | Surface Skin Temperature | 2010 MODIS/Aqua Land Surface Temperature and Emissivity (LST/E) product |
| U10 | 10-meters wind at U direction | 2010 PRD 10-Meters Wind Speed |
| V10 | 10-meters wind at V direction | |
| RAINC | Accumulated total cumulus precipitation | 2010 PRD Precipitation |
| RAINNC | Accumulated total grid scale precipitation | |





| RH2 | 2-meters relative humidity | 2010 PRD Relative Humidity |
|---|---|---|

### 2.4 Model Evaluation Methods

For each of the meteorological variables, we measured not only the spatial variation range and median values of the modelled and observed variable but also the difference values between the modelled variable and its corresponding observed one at each spatial-temporal epoch. Then, instead of merely checking the average and standard deviation of the bias, we respectively analyzed the probability density function (PDF) of modelled and observed variables, the temporal-spatial variation of the modelled and observed variable, and the probability of the difference values between modelled variable and its corresponding observed one.

Firstly, we quantified the extent of overlap between the observed and modelled variables' PDFs by using the Perkins Skill Score (PSS). PSS ranges from 0 to 1, while 1 indicates perfect modelling and 0 indicates the worst modelling. Secondly, since temporal and spatial variations are vital for the meteorological phenomenon, we specifically compared the modelled variables and their corresponding observed ones by using the graphic of temporal comparison of spatial variation which displays the annual variation of the modelled and observed variable's spatial variation (range and median values) at the temporal dimension. In doing so, we got a direct sense of whether the modelled results replicated the temporal and spatial features in the observations. Thirdly, we visualized and analyzed the general magnitude and distribution of the bias by generating the monthly PDF of the difference values between the modelled variable and its corresponding observed one. Simple descriptive statistics such as the mean and standard deviation can be misled by outliers and thus miss the actual patterns in the data, while PDF visualizes the entire data distribution. With visualized PDFs, we can understand the empirical distribution of the simulation bias and notice outliers directly if any, which may shed light for further model calibrations. Finally, we further hybrid PDF into the investigation of temporal variation by visualizing and comparing the PDF of the difference values in each month of the year.

## 3 Technical Preparation

### 3.1 Model Setup

A telescoping nests' structure with four nested domains which are centered at 22°39′30″ N, 114°11′30″, was set up as the horizontal domain baseline configuration in this study. Moreover, the same set of eta levels with 51 members was used in each horizontal domain. Furthermore, there were some physics components in the model, and each component had some different schemes for choosing. Table 3 shows the scheme chosen for each component.

**Table 3: Physics Components' Schemes.**

| Component | Scheme |
|---|---|
| Cumulus | New Simplified Arakawa-Schubert |
| Microphysics | WDM5 |
| Radiation | RRTMG |
| Planetary Boundary Layer | Bougeault–Lacarrere |
| Surface Layer | Revised MM5 |
| Land Surface Model | Noah LSM |
| Urban Canopy Model | Single-layer |





### 3.2 Data Preparation

Firstly, the 2010 NCEP FNL (Final) Operational Global Analysis Dataset (1-degree grid spatial resolution and 6-hourly temporal resolution) was used as the Gridded Data in this study. Secondly, the Completed Dataset of WRF Preprocessing System (WPS) Geographical Input Data was used as the Static Geographical Dataset in this study. Thirdly, the 2010 PRD

Urban Land Surface Dataset, whose major sets of data include the Vegetation Coverage Dataset, Building Dataset, Artificial Impervious Area Dataset, and Anthropogenic Heat Dataset, was specially developed for refining the WRF primary data.

### 3.3 Primary Data Processing

Firstly, the primary data included the interpolated geo-data files, the intermediate format meteorological data files, the horizontally interpolated meteorological data files, the initial condition data files, and the lateral boundary condition data

files. Secondly, two primary data processing software packages (geo_data_refinement processing package and wrf_input_refinement processing package) were developed for extracting the urban land surface attributes from the 2010 PRD Urban Land Surface Dataset and revising the corresponding fields of the related primary data files with these attributes.

### 4 Model Evaluation

### 4.1 Evaluation of the 2-Meters Air Temperature

Firstly, as shown in Figure 1, there are 57 temperature observations with 2-meters air temperature data in domain 4, out of which 23 are located in non-urban areas and 34 in urban areas. Secondly, as shown in Figure 2, the monthly PSS of 2-meters air temperature ranges from a minimum of 0.595 in July to a maximum of 0.886 in January and has an annual mean value of 0.724. This makes it clear that the model captured the PDF for the observed air temperature at least about 60% in a month and over 72% in a year. Thirdly, Figure 3 shows monthly comparisons between the observed and the modelled 2-meters air

temperatures' spatial variation range and median values at 2:00, 8:00, 14:00, and 20:00. It is evident that the modelled air temperatures always have similar behaviour in temporal-spatial variation with the observed ones. Fourthly, Figure 4 shows the diurnal variations of observed, modelled air temperatures' median and spatial variation range in each month. As is evident in Figure 4, both the median and the range of the 2-meters modelled air temperature have the same diurnal variation pattern as that of its corresponding observed ones in each month, although there are differences between the modelled ones

and the corresponding observed ones. Finally, Figure 5 shows that the PDF of differences that exist between each value of each month's time series of modelled grid air temperatures and its corresponding observed ones. As shown in Figure 5, the probability of 3-degrees bias interval (the absolute value of the difference between modelled surface temperature and its corresponding observed one is 3 degrees) in a month varies from 64% to 91% and has an annual mean probability of this interval of 78%.

To sum up, the model produces quite a good simulation of 2-meters air temperature with annual mean PSS of 0.724. It also captures the behaviors of monthly and diurnal variation of observed 2-meters air temperatures. However, as shown in Figure 2, the modelled distribution shifts to low temperature in the period of June to October (summertime in the research area). Actually, as shown in Figure 5, the differences between the modelled 2-meters air temperatures and their corresponding observed ones exist in the whole year. In reality, the difference includes not only the modelling bias but also a natural gap

between a 1-km grid spatial average value and a value of a point located in this grid. Moreover, the observation always locates in an open area, and thus, the observed 2-meters air temperature is the temperature of a point in the open area. The modelled 2-meters air temperature is a mean temperature of a 1-km grid which always includes some vegetation covered areas. It is a common sense that the point air temperature in the open area is always higher than its corresponding 1-km grid mean air temperature in the summertime.





### 4.2 Evaluation of Surface Temperature

Firstly, the MODIS/Aqua Land Surface Temperature and Emissivity (LST/E) product (Short name: MYD11A1) provided by the U.S. Geological Survey (USGS) is used for evaluation. This product includes a 1-km horizontal resolution grid surface temperature at around 2:00 and 14:00 (Beijing time) per day. It also has a quality control attribute for each surface temperature record to identify the level of data quality. Such quality control attribute is used for filtering the records that lack quality and differ from the corresponding modelled value by over 5 degrees. Secondly, Figure 6 and Figure 7 show that the monthly PSS of modelled surface temperatures ranges from 0.629 to 0.794 at 2:00 and from 0.479 to 0.777 for modelled at 14:00 respectively. Moreover, the annual mean PSS of modelled temperatures at 2:00 and 14:00 is 0.702 and 0.623 respectively. Accordingly, both modelled surface temperatures at 2:00 and 14:00 are seen as quite a good expounding in MODIS surface temperature with a PSS of over 0.6. Thirdly, as shown in Figure 8, the modelled surface temperatures have the same annual variation pattern as their corresponding MODIS one irrespective of whether they are measured at 2:00 or 14:00. Fourthly, as shown in Figure 9 and Figure 10, both the modelled surface temperatures and their corresponding observations from MODIS have the same pattern in which the surface temperature is higher in the urban areas than in non-urban areas irrespective of the time at which it is measured. Therefore, the model successfully captures the urban climatological behavior in which the surface temperature is higher in the urban areas than in non-urban areas. Finally, as shown in Figure 11, the monthly probabilities of 3-degrees bias interval (the absolute value of the difference between modelled surface temperature and its corresponding MODIS one is 3 degrees) at 14:00 ranges between 54% and 84% and has quite a high annual mean value of 73%. Moreover, Figure 12 also shows that the probabilities of a 3-degree bias interval at 2:00 ranges from 69% to 98% and has a really high annual mean value of 87%.

To sum up, the modelled 14:00 and 2:00 surface temperatures represent the corresponding MODIS ones with an acceptable PSS. Moreover, the modelled surface temperatures also have the same annual variations and the same urban climatological patterns as that of the MODIS ones. However, as shown in Figure 11 and Figure 12, the difference between the modelled surface temperature and its corresponding MODIS one is noticeable in some grids. An analysis which was conducted to the MYD11A1 dataset finds that there are many grids whose quality was not evaluated in the MYD11A1 dataset and accordingly, it is highly possible that this difference includes an observation bias. Moreover, due to the difference between the temporal coverages of the model outcome and its corresponding observation from MODIS, the observed difference also includes a bias introduced by the difference in measured time. Furthermore, the resampling operation on the MODIS dataset also causes a technical bias in some grids.

### 4.3 Evaluation of the 10-Meters Wind Speed

Firstly, as shown in Figure 13, there are 62 observations with 10-meters wind speed data in domain 4, out of which 26 are located in non-urban areas and 37 in urban areas. Secondly, as shown in Figure 14, the monthly Perkins Skill Score of modelled 10-meters wind speed ranges between 0.482 and 0.802 and has an acceptable annual mean value of 0.660. Thirdly, Figure 15 shows monthly comparisons between the spatial variation range and median values of the observed and the modelled 10-meters wind speeds at 8:00, 14:00, 20:00, and 2:00. It is evident that the modelled 10-meters wind speed always has a similar behaviour of temporal-spatial variation with the observed ones. Finally, as shown in Figure 16, the monthly probabilities of 3 m/s bias interval (the absolute value of the difference between modelled wind speed and its corresponding observed one is 3 m/s) range between 61% and 83%.

To sum up, we provide a well-grounded conclusion that the model simulates 10-meters wind speed with acceptable PSS. The modelled ones of 10-meters wind speed also have the same annual variation as that of the observed ones. However, Figure 14 shows that the modelled distribution shifts to high speed. The difference in the speed of modelled 10-meters wind and its corresponding observed one is not entirely caused by the model bias. The observation altitude of the modelled 10-meters





wind is different from its corresponding observed one. The modelled outcomes measure the upper air movement of the urban canopy, but the observations measure the air movement inside the canopy. The locations of modelled and observed air movements concerning the canopy would cause a natural gap between the modelled and observed values. Moreover, this difference also includes a natural gap between a 1km-grid spatial average value and a value of a point located in this grid.

## 4.4 Evaluation of Precipitation

Firstly, as shown in Figure 17, there are 64 observations with precipitation data in domain 4, out of which 33 are located in non-urban areas and 31 in urban areas. Secondly, as shown in Figure 18, the monthly PSS of modelled precipitation ranges between 0.444 and 0.747 and has an annual mean value of 0.579. Thirdly, Figure 19 shows monthly comparisons between the observed and the modelled precipitation' spatial variation range and median values at 8:00, 14:00, 20:00, and 2:00. It is evident that the modelled ones of precipitation always have similar behaviour of spatial-temporal variation with the observed ones. Finally, as shown in Figure 20, the model simulated precipitation with an accuracy in which the monthly probabilities of the 3-mm bias interval (the absolute value of the difference between modelled precipitation and its observed one is 3 mm) range between 39% and 89% and have an acceptable annual mean value of 67%.

Based on the comparison of experiments and observations concerning the modelled and observed measurements of precipitation, we provide a well-grounded conclusion that the model simulates precipitation with an acceptable PSS. Moreover, the modelled precipitations also have the same annual variation as that of the observed ones. However, Figure 20 shows that the probability of 3-mm bias intervals is quite low in some months; for example, the one was 39%, 50%, and 53% in June, September, and May respectively. The modelled precipitations deviated from its corresponding observed ones in these three months.

## 4.5 Evaluation of Relative Humidity

Firstly, as shown in Figure 21, there are 24 observations with relative humidity data in domain 4, out of which nine are in non-urban areas and 15 in urban areas. Secondly, as shown in Figure 22, the monthly PSS of the modelled relative humidity ranges between 0.525 and 0.786 and has an annual mean value of 0.673. Thirdly, Figure 23 shows the monthly comparisons between observed and modelled air relative humidity values' spatial variation range and median values at 8:00, 14:00, 20:00, and 2:00. It is apparent that the modelled values always have similar behaviour in spatial-temporal variation with the observed ones, although all modelled median values are higher than the corresponding observed ones. Finally, as shown in Figure 24, the model simulates the relative humidity with quite a good accuracy in which the monthly probabilities of the 20% bias interval (the absolute value of the difference between modelled precipitation and its observed one is 20%) range between 77% and 96% and have a high annual mean value of 91%.

To sum up, the model simulates the relative humidity with an acceptable PSS and accuracy. Moreover, it also simulates the monthly variation pattern of relative humidity appropriately.

## 5 Discussions and Conclusions

### 5.1 Model evaluation using observations

We need more sophisticated model evaluation methods for better comparisons between model outcomes and observations to serve as partial support for the reliability of urban climate simulations and any conclusions based on the simulation results. A model can simulate a resonance with the natural system (Oreskes et al. 1994), and accordingly, a climate simulation should aim at modelling the temporal and spatial meteorological features of climate. Therefore, a model evaluation should aim at assessing the similarity of temporal and spatial features between the modelled results and observations. In this study, the PSS





was used for assessing the similarity quantificationally, and the graphic of temporal comparison of spatial variation was used for assessing the similarity qualitatively. The satisfied quality of simulation was evaluated by the acceptable annual values of PSS of modelled variables and the similar behaviors of modelled variables with their corresponding observed ones shown in the graphic of temporal comparison of spatial variation figures of modelled variables.

Utilizing the proposed model evaluation methods, evaluation results in this case study indicate that this atmospheric model appropriately portrayed the annual variations in the climatological patterns of air temperature, surface temperature, 10-meters wind speed, and air relative humidity. We observe that the simulation model captured similar temporal and spatial meteorological features of urban climate. From a quantitative perspective, the model achieved at least an acceptable PSS and accuracy in the simulations of 2-meters air temperature, surface temperature, 10-meters wind speed, precipitation, and air

relative humidity, which means that the simulation results are acceptable approximations of the observations. Apparently, according to the above evaluations, the proposed simulation model in our case study is sufficiently reliable in reproducing meteorological features of urban climate at the 1-km spatial resolution.

The good match in our study or any other study, between the model outcomes and observations can only support that the simulation results are acceptable approximations of the observations in the specific spatial-temporal coverages in respective

studies. These comparisons are inadequate for model 'verification' or 'validation'. Returning to the philosophical basis, terminologies "verification" and "validation" imply the confirmation of truth and legitimacy respectively (Oreskes et al., 1994). We get observations of meteorological characters from monitoring stations, and that is why the observations come in points and suffer from frequent missing data. Therefore, it is common that the spatial-temporal coverage of the observations can only partially match that of the modelling outcomes, which can be proved by the model evaluation process regarding air

temperature, surface temperature and other factors mentioned above. A good match between a model outcome and its corresponding observation at specific locations is no guarantee of a good match at other locations. Similarly, a good match between the model outcomes and the corresponding observations for a historical period is no guarantee of a good match in the future. Moreover, a good match between the model outcomes and corresponding observations for a limited spatial-temporal range does not guarantee that the model is free from initial and uncertainties. Consequently, even a complete match

between the observations and model outcomes does not ensure a successful verification and validation of the modelling system, let alone an incomplete match in practice (Oreskes et al., 1994). Although it is impossible to verify or validate a model, it is feasible to evaluate model outcomes with the observation data using sophisticated spatial and temporal comparisons, since a model can represent a natural system accurately to some extent (Oreskes et al. 1994) and accordingly, evaluating the model with the observation seems to be the best way we currently have to access the performance of a model,

but we should always be aware of its imperfectness.

### 5.2 The Natural Gap, Observation bias, and Model bias

The model evaluation is not complete even a complete match between the observations and model outcomes does not ensure a successful verification and validation of the modelling system, let alone an incomplete match in practice (Oreskes et al., 1994). Observations are probably the best reference we get to evaluate the simulation results, but that does not mean

observations are perfect for such an evaluation. The comparison between the model outcome and observations alone does not make a complete model evaluation since it does not rule out the natural gap, observation bias, and model bias.

The natural gap refers to the fact that model outcomes from the simulation models are average values of a grid, while the observations are point-based which only measures the meteorological conditions around the location of the monitoring station. Comparing the average value within a spatial area, the size of which ranges from $0.25$ km$^2$ to over $100$ km$^2$, with

point-based observations is problematic for two main reasons. 1) The average value in a grid is calculated under the assumption that the grid is homogeneous, which is usually not true especially when detailed urban morphology is





considered, and so the average value is usually lower than that of point-based observations; 2) the point-based observations are likely to be significantly affected by the surrounding environment of the monitoring station, which does not reflect the meteorological condition in the same area. Therefore, the comparison between modelled outcomes and observations has biases, although it is usually the only model evaluation approach we get so far. The only exception is using the observations

from remotely sensed imagery, for example, we used the land surface temperature product from MODIS/Aqua to evaluate the modelled temperature of the surface skin. However, there are many grids whose quality have not been evaluated in MODIS/Aqua Land Surface Temperature Product, and accordingly, the difference between the modelled temperature of the surface skin of a grid and its corresponding one in MODIS/Aqua Land Surface Temperature Product includes an observation bias highly possible.

The model bias refers to the uncertainty caused by differences between the actual atmospheric physical processes and the approximations in the model (Skamarock et al., 2005, 2008). The atmospheric model produces the fine atmospheric features which do not exist in the original meteorological data. An atmospheric model can represent a natural atmospheric system accurately to some extent rather than entirely. The simulation models are supposed to include many more complex atmospheric physical processes to explain meteorological states with high spatial and temporal resolutions, but many of them

have to be omitted or empirically approximated due to limitations in knowledge and computational efficiency. This situation is unusually severe for high-resolution climate simulations. Given the complexity of simulation models, estimating error propagation in these models are difficult, and thus model evaluation becomes the only quality control of simulation results, especially for high-resolution urban climate simulations which are more sensitive to the inadequacies of the atmospheric model, inappropriate configuration of the modelling system (Warner, 2011), and the quality of input data.

**5.3 Conclusions**

In conclusion, we emphasize in this paper that model evaluation is necessary and usually the only process that guarantees the reliability of simulation outcomes, and so utilizing a sophisticated model evaluation process to reach an acceptable agreement between the simulated and observed meteorological variables should be the premise of any conclusion drawn from the modelling results. The emerging high-resolution urban climate simulation models are especially sensitive to

possible initial and model uncertainties. In this vein, we proposed a sophisticated model evaluation framework that examines not only the matches between the spatial-temporal patterns of the modelled and observed variables but also the statistical distribution of the difference between the modelled variables and their corresponding observations. Moreover, the proposed method utilized PSS to statistically quantifies the extent of overlap between the PDFs of modelled variables and their corresponding observations, which, we argue, was a more informative and useful indicator for the quality of modelling

outcomes compared to existing metrics such as residuals and correlations. By doing so, we hope to provide more capable tools that improve the quality control in future researches using numerical meteorological simulations, especially high-resolution urban climate simulations. We also intend to raise the awareness and attention over model evaluation methods within the modelling community, since new findings without sophisticated understanding, control of model uncertainties and systematic assessments of model outcomes may be scientifically misleading. Finally, we reminded that the modeller should

be cautious to conclude a quantitative finding because it is impossible to identify the natural gap, observation bias, and model bias in the difference between observations and its corresponding modelled results.



*Code availability*. Information on the availability of source codes used in this study is tabulated below.

| Source codes | Availability |
|---|---|
| WRF Model 3.7.1 | These source codes are publicly available at |
| WRF Pre-Processing System (WPS) 3.7.1 | http://www2.mmm.ucar.edu/wrf/users/download/get_source.html |
| namelist.wps | These source codes are available upon request from the corresponding author. |
| namelist.input | |
| Changes in the programs of WRF for inputting the 2D anthropogenic sensible and latent heat data | |
| geo_data_refinement processing package | |
| wrf_input_refinement processing package | |

*Data availability*. Information on the availability of data used in this paper is tabulated below.

| Data | Availability |
|---|---|
| 2010 NCEP FNL (Final) Operational Global Analysis Dataset | This dataset is publicly available at https://rda.ucar.edu/datasets/ds083.2/ |
| Completed Dataset of WRF Preprocessing System (WPS) Geographical Input Data | This dataset is publicly available at http://www2.mmm.ucar.edu/wrf/users/download/get_sources_wps_geog.html |
| 2010 PRD Observation Locations | These datasets are available upon request from the corresponding author. |
| 2010 PRD Urban Land Surface Dataset | |
| 2010 PRD 2-Meters Air Temperature | |
| 2010 PRD 10-Meters Wind Speed | |
| 2010 PRD Precipitation | |
| 2010 PRD Relative Humidity | |
| 2010 MODIS/Aqua Land Surface Temperature and Emissivity (LST/E) product | This dataset is publicly available at https://lpdaac.usgs.gov/dataset_discovery/modis/modis_products_table/myd11a1_v006 |
| Modelling Variables for Model Evaluation (T2, TSK, U10, V10, RAINC, RAINNC, RH2) | This dataset is available upon request from the corresponding author. |

*Competing interests*. The authors declare that they have no conflict of interest.

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



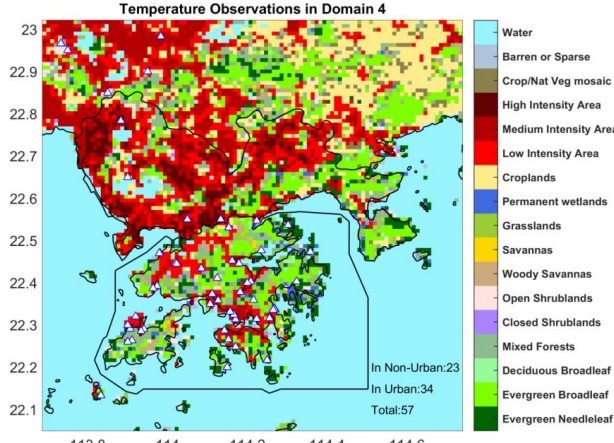

Figure 1: Temperature Observations in Domain 4.

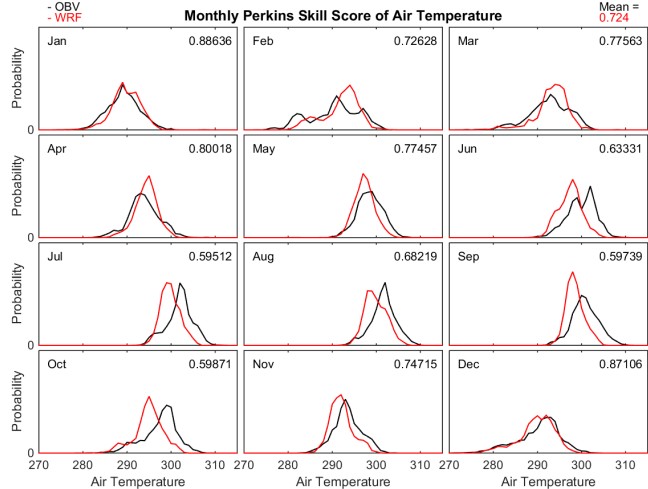

Figure 2: Monthly PSS of 2-Meters Air Temperature.

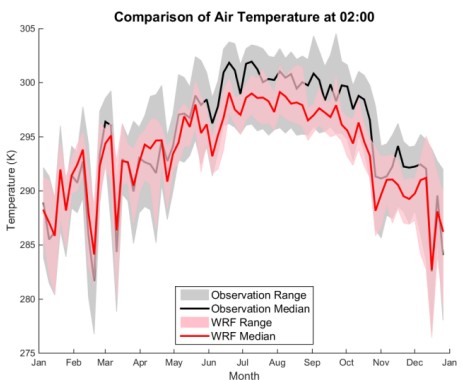
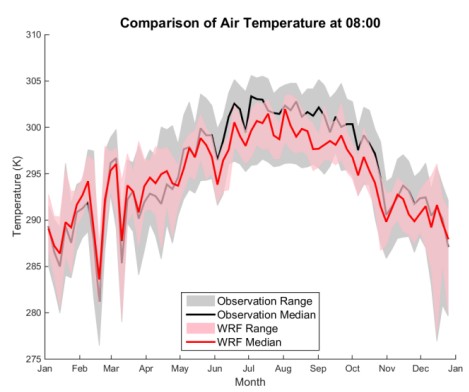



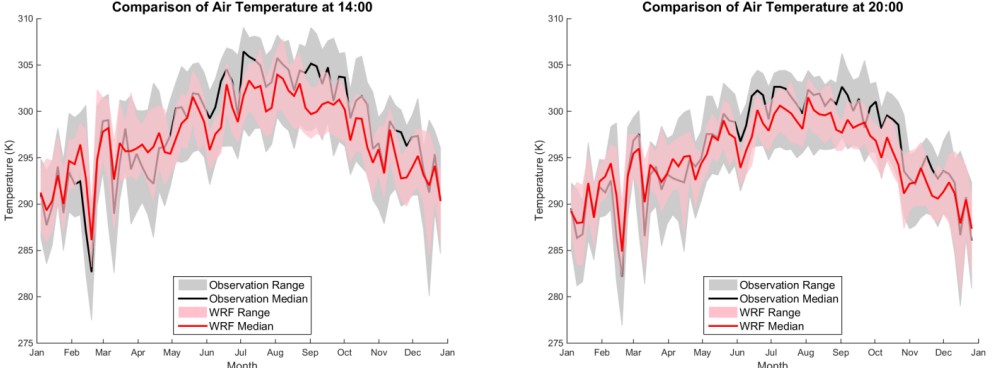

**Figure 3: Comparison of 2-Meters Air Temperature at 2:00, 8:00, 14:00, and 20:00.**

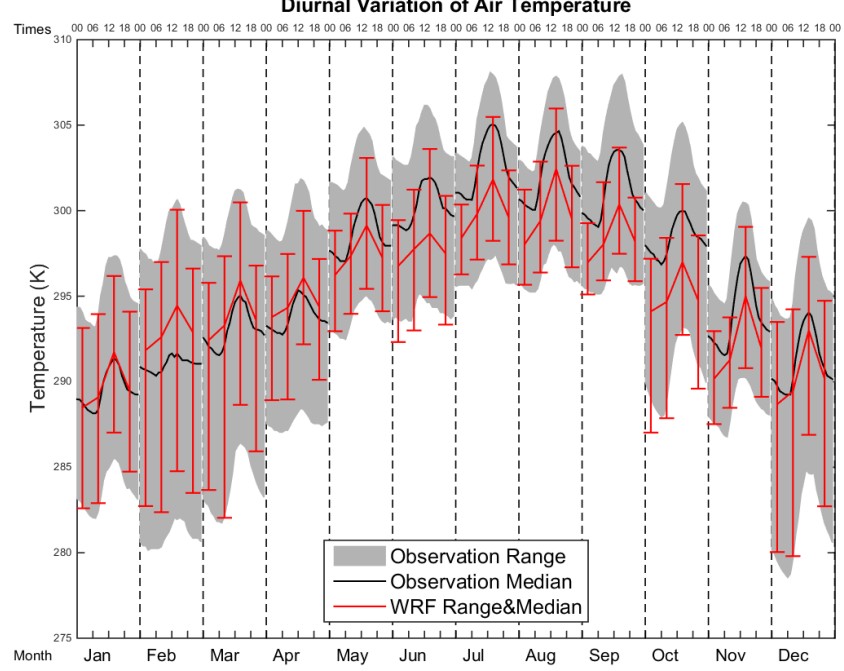

**Figure 4: Diurnal Variation of 2-Meters Air Temperature.**




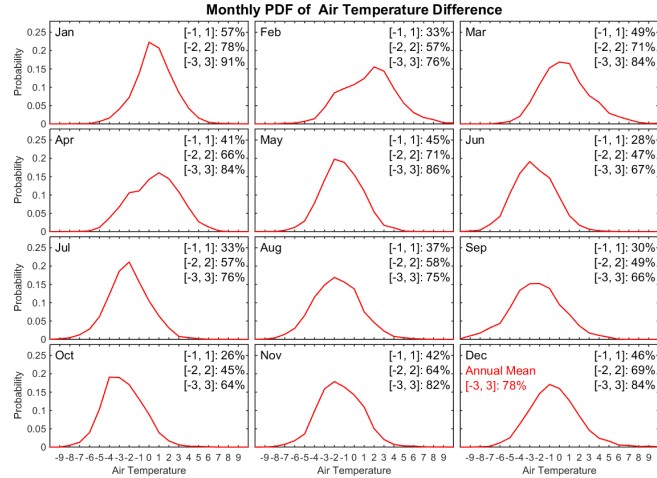

**Figure 5: Monthly PDF of 2-Meters Air Temperature Difference.**

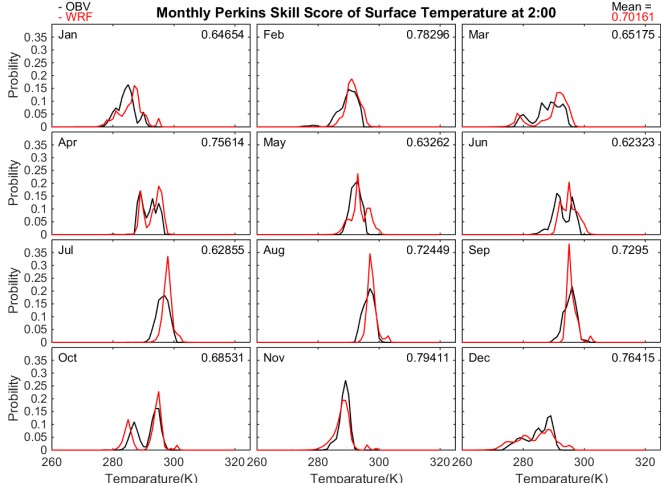

**Figure 6: Monthly PSS of Surface Temperature at 2:00.**

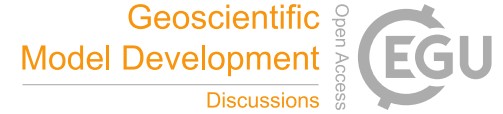

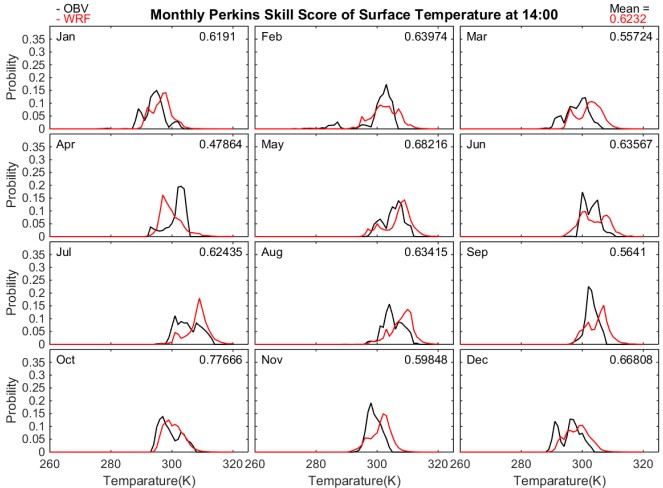

**Figure 7: Monthly PSS of Surface Temperature at 14:00.**

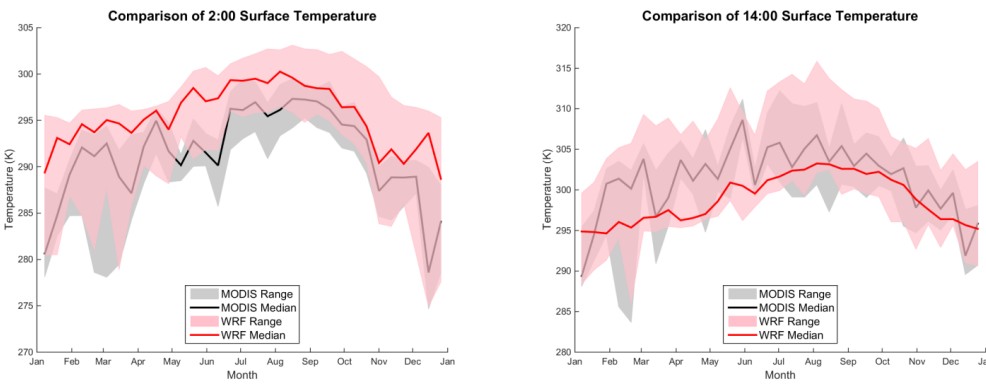

**Figure 8: Comparison of Modelled Surface Temperatures with its Corresponding MODIS Ones at 2:00 and 14:00.**

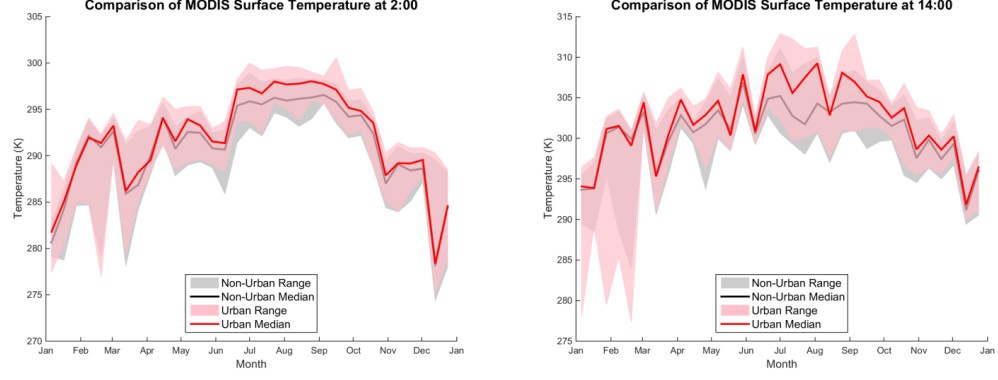

**Figure 9: Comparison of MODIS Surface Temperatures in Urban Area with the Ones in the Non-Urban area at 2:00 and 14:00.**





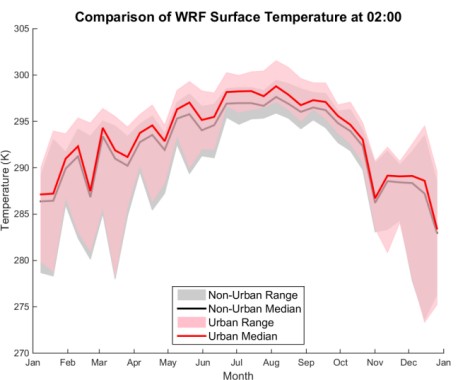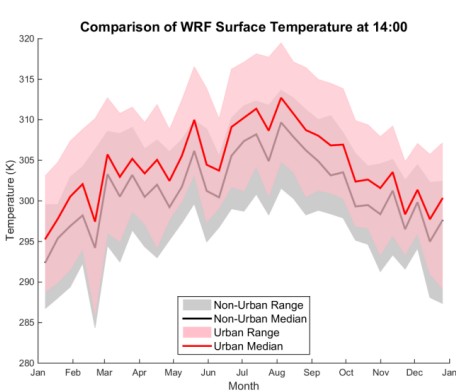

**Figure 10: Comparison of Modelled Surface Temperatures in Urban Area with the Ones in the Non-Urban area at 2:00 and 14:00.**

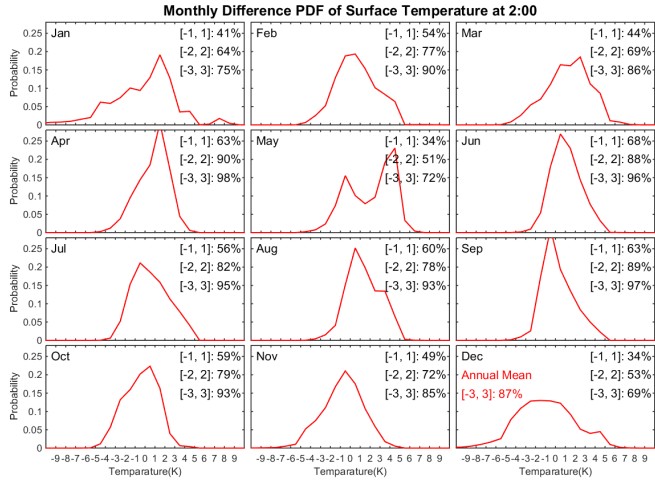

**Figure 11: Monthly PDF of 2:00 Surface Temperature Difference.**

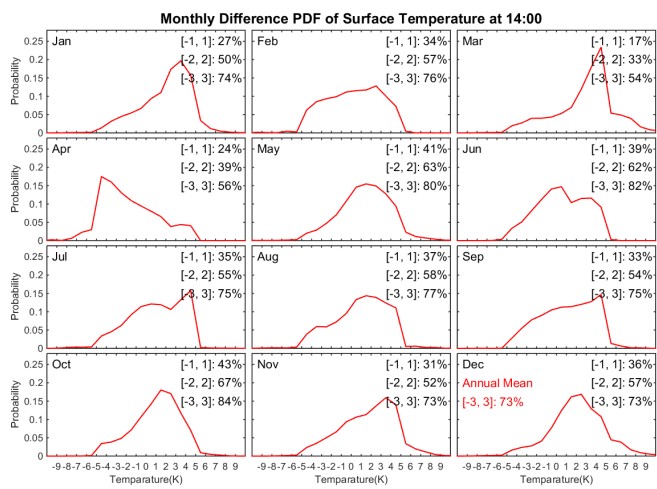

**Figure 12: Monthly PDF of 14:00 Surface Temperature Difference.**





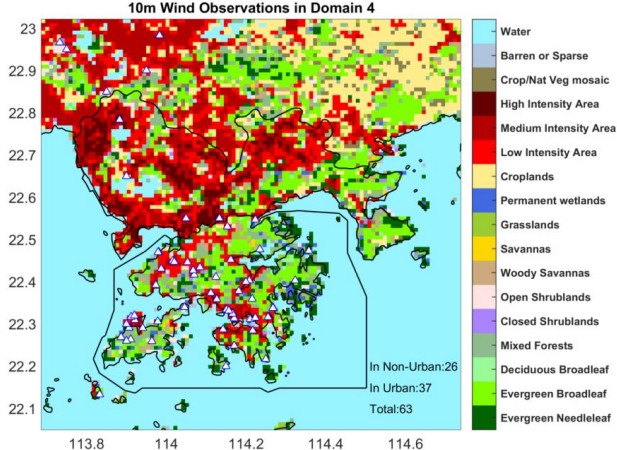

**Figure 13: The 10-Meters Wind Observations in Domain 4.**

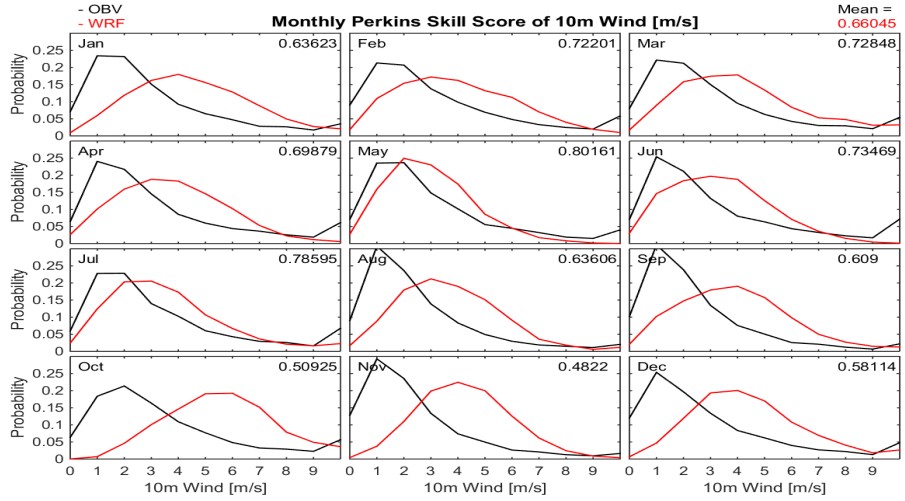

**Figure 14: Monthly PSS of 10-Meters Wind Speed.**

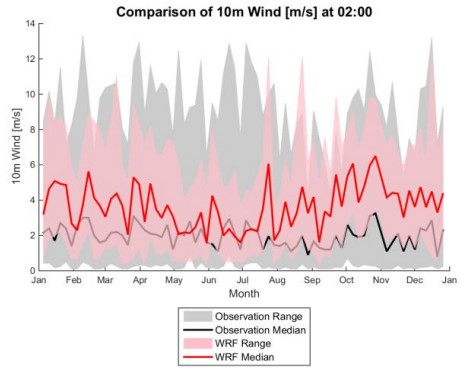

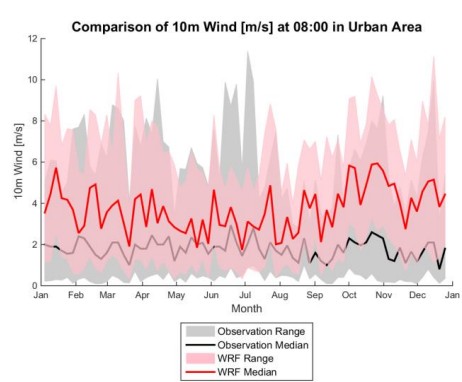

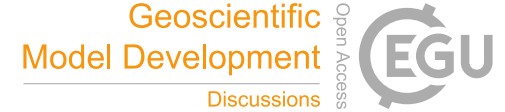



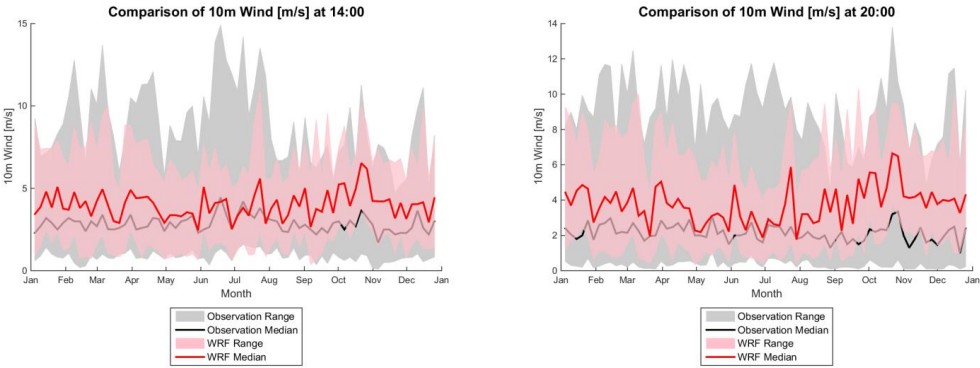

**Figure 15: Comparison of Modelled 10-Meters Wind Speed with its Corresponding Observed Ones at 2:00, 8:00, 14:00 and 20:00.**

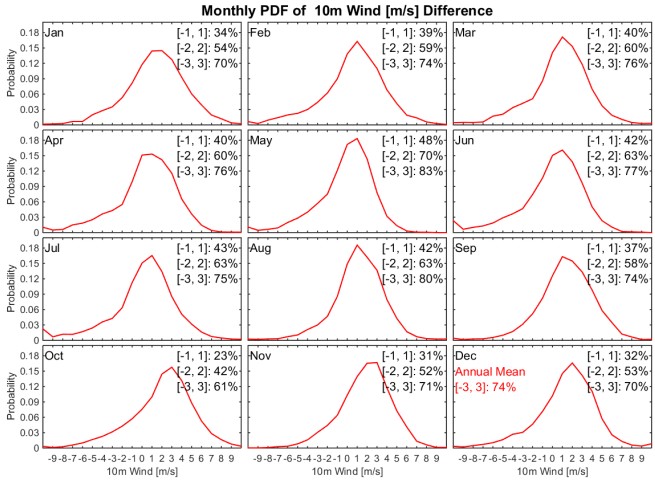

**Figure 16: Monthly PDF of 10-Meters Wind Speed Difference.**

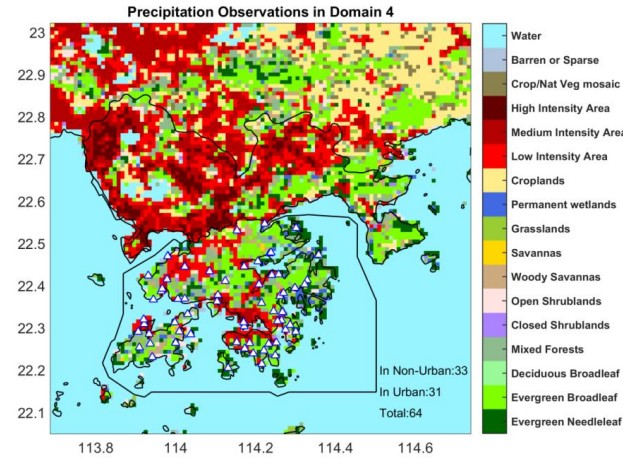

**Figure 17: Precipitation Observations in domain 4.**



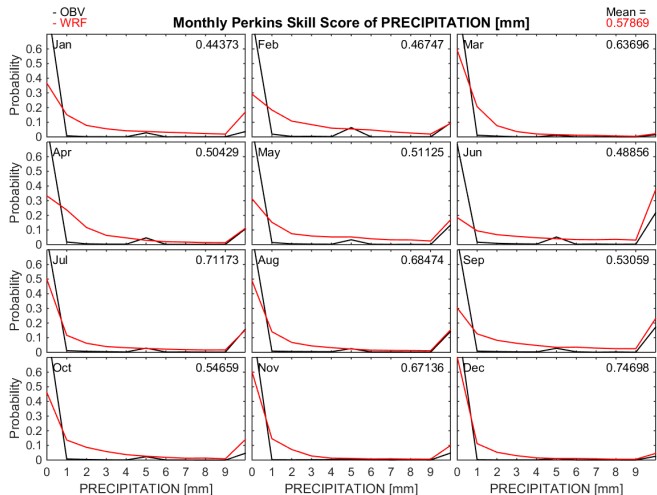

**Figure 18: Monthly PSS of Precipitation.**

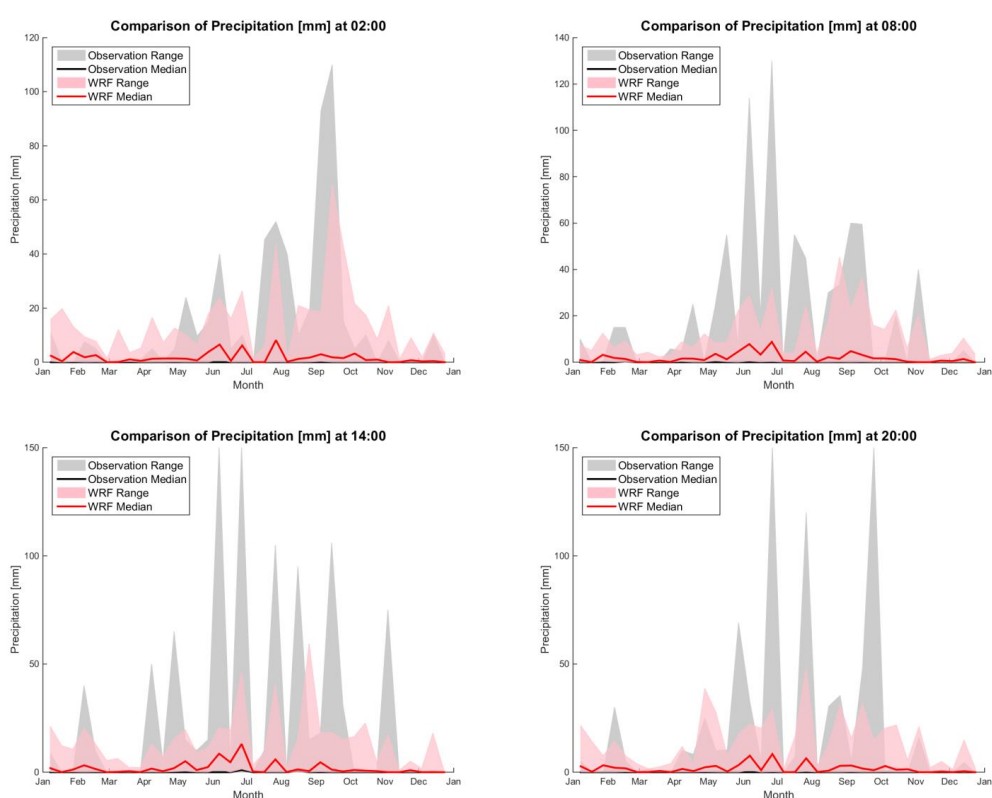

5    **Figure 19: Comparison of Modelled Precipitations with its Corresponding Observed Ones at 2:00, 8:00, 14:00, and 20:00.**




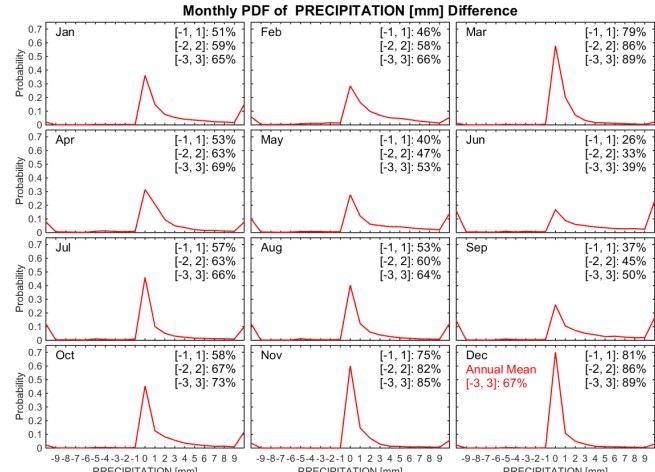

**Figure 20: Monthly PDF of Precipitation Difference.**

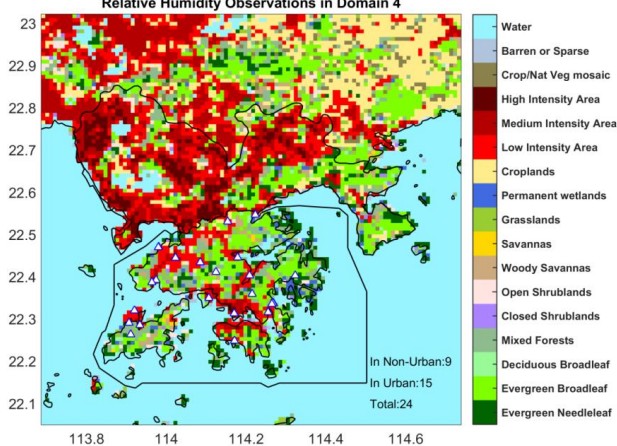

**Figure 21: Relative Humidity Observations in domain 4.**

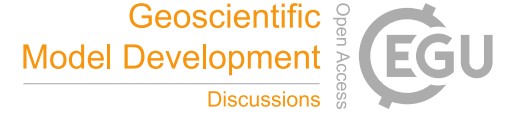



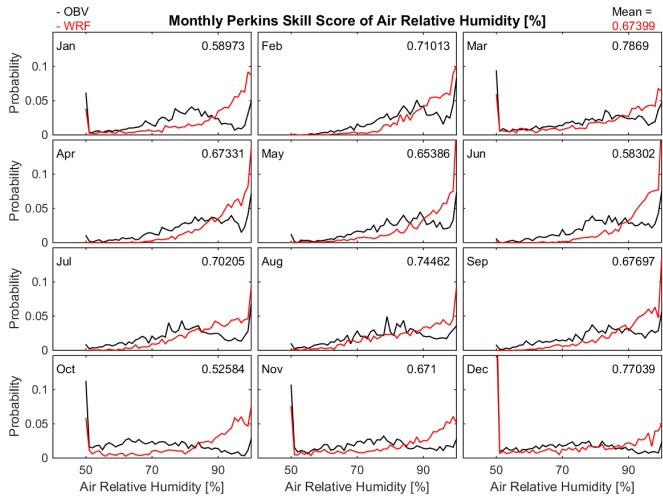

**Figure 22: Monthly PSS of Relative Humidity.**

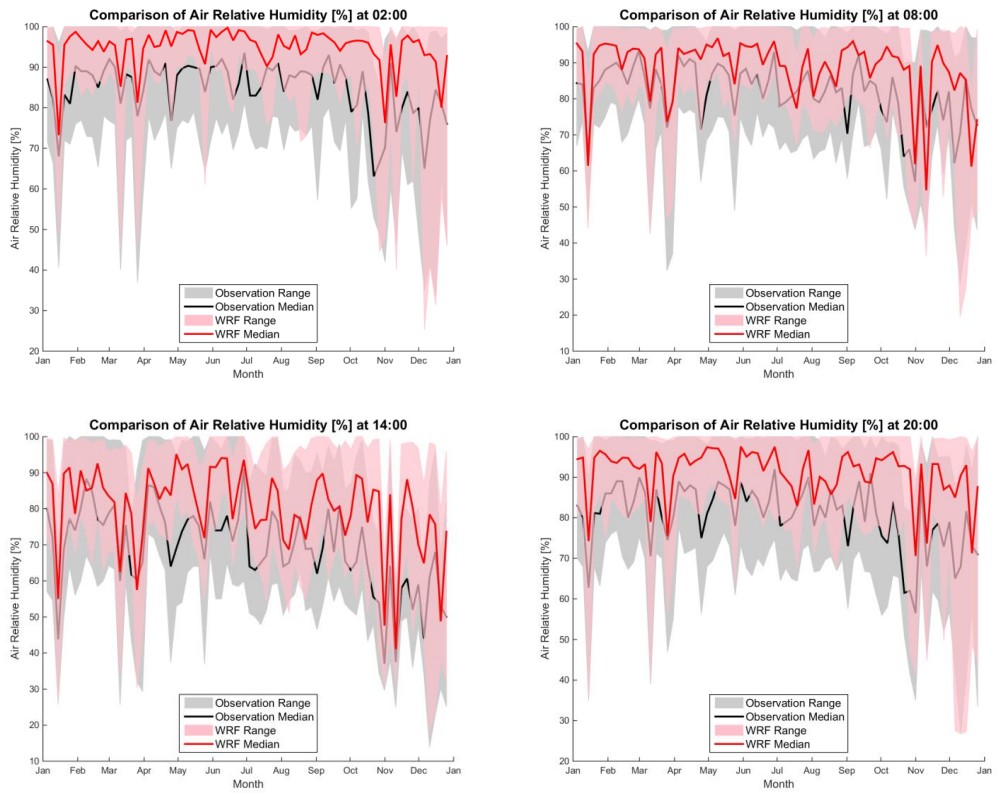

5    **Figure 23: Comparison of Modelled Relative Humidity with its Corresponding Observed One at 2:00, 8:00, 14:00, and 20:00.**