# Peer review of "Model evaluation of high-resolution urban climate simulations: using WRF/Noah LSM/SLUCM model (Version 3.7.1) as a case study"

_Geoscientific Model Development, 2018_

## Short Comment (SC1) · 5 Nov 2018

Dear authors,

please note, that the title of a GMD manuscript should state the model name (or its acronym) and a version number. This is also important information in the case of an evaluation, as different versions might perform differently for the same evaluation procedure. Therefore please add a version number of WRF etc. to the title of your manuscript upon revision.

Best regards,

Astrid Kerkweg (executive Editor)

---

## Author Comment (AC1) · 5 Nov 2018

Dear A. Kerkweg,

Thank you very much for your gentle reminding. The WRF ARW modelling system version 3.7.1 was used in this study. We will revise the title for adding the version number after the interactive discussion.

Best,

Sincerely,

Zhiqiang Li, Ph.D. in Earth System and Geo-Science PaterLee@link.cuhk.edu.hk Insti-

tute of Space and Earth Information Science, The Chinese University of Hong Kong

---

## Referee Comment (RC1) · Anonymous Referee #1 · 25 Nov 2018

Review for 'Model evaluation of high-resolution urban climate simulations: using WRF ARW/LSM/SLUCM model as a case study' by Zhiqiang Li et al.

General comments: This study evaluates performance of the WRF model in terms of high-resolution urban climate modelling over an area encompassing two big cities, Shenzhen and Hong Kong. The chosen area of Shenzhen is heavily urbanized but only a small part of Hong Kong is urbanized. Perkins skill score is used as a major evaluation method throughout the evaluation. The authors argue that their study has proposed a methodological framework for evaluating model performance in high-resolution urban climate simulation. I think this work is useful and has provided some

information about high-resolution urban climate modelling applied to south China. I very much appreciate the authors' efforts to pursue this kind of modelling work. However, I feel that the manuscript in the current form cannot be accepted for publication. At a minimum, I would suggest some necessary revisions to make the paper publishable in the journal. But to engender a stronger paper, I feel that more extensive work might have to be done. I will leave it to the editor to decide whether such extensive work is required.

Major comments: 1) The introduction should be reformulated with greater care. The authors should survey the literature more thoroughly. Only a few papers are mentioned in the introductory section. I suggest the authors give a good overview of the existing studies on the topic, and point out the limitations of the past studies and challenges/constrains. Identifying a gap or proposing a new method as well as outlining the contributions of the study is also helpful.

2) The data and methodology section should be structured in a more logical way. I think the authors could place model description and experiment/model setup before evaluation method. Overall, both section 2 and 3 are a bit confusing. The introduction of the model is lacking. The authors should clearly articulate what has been done and how it has been done. This can aid the readers in understanding the experiment setup/design.

3) In section 2.1, more details about the new dataset developed by the authors should be offered. The reasons for focusing on the simulations in the year of 2010 should be discussed. In section 2.2, more details should be provided as to the four-day segment simulations. Did the model read in restart files every four days to continue the simulation? How may a different simulation strategy affect the modelling results? In section 2.3, instead of just giving two tables, I think more detailed descriptions of the data should be given. How are the comparisons between model output (grid points) and observations (stations) made? Representativeness of the observations and potential biases should be discussed. The authors should also indicate the reasons for

choosing evaluation variables.

4) In section 2.4, no references are cited regarding the Perkins skill score. Is this a suitable method for this study? There should at least be some discussion. Authors should also discuss whether this method is suitable for all the variables evaluated in the study.

5) In section 3, choosing of the parameterization schemes needs discussion.

6) I think the authors should tune down many of their arguments throughout the paper to avoid overstating (e.g., P2L25-26). For example, I don't see any strong methodological framework being discussed and described in the text.

7) I have the impression that the authors have been too obsessed with 'good results' when evaluating the model's performance. Discussing 'good results' and 'bad results' at the same time, in my opinion, is fair. It's perhaps more important to identify areas for improvements.

8) The structure and writing are too repetitive in section 4. This is also true for the figures. The number of figures may be reduced. While the focus of the paper as stated in the paper is on the urban climate simulation, evaluation seems to be applied to also the vast rural regions. The authors should clarify this. I suggest the authors focus on the most important aspects of the urban climate simulation. I would suggest some points (see following) for the authors to consider and they should further develop a better evaluation framework.

-Some basic ability of the model such as spatial distribution temperature/precipitation and diurnal cycles of temperature must be assessed.

- The weather and climate variability in the study area is strongly associated with the monsoon flow. So the investigation of the simulation of precipitation and temperature is rather important. Both the spatial distribution (not found in any of the figures in the paper) and temporal variability should be considered. In particular, the authors

may identify some strong urbanization impacts on the precipitation (e.g., precipitation maxima) and temperature (e.g., urban heat island). The model's ability to capture these effects is essential. In addition, simulation of sea breeze, wind distribution, boundary layer variability, and stability of the atmosphere should be examined. The impact of urbanization on the air quality may also be discussed.

- The evaluation can be done separately for different seasons. The evaluation should focus on the most important aspects of urban climate/weather.

- The scientific value can be enhanced if the authors can demonstrate how the model behaves in simulating the extreme precipitation events or heat wave/cold surge events, and How and to what extent these events may be related to the urbanization.

- The model's performance between different regions in the study area and between rural and urban regions can also be compared.

9) The figures can be better designed and drawn. Captions of the figures should provide more information. The language could also be improved.

Minor comments: The authors should check carefully the use of words and sentences throughout the paper. I suggest some serious edits/revisions. I list only some of the examples. P1L15: add 'have' before paid. P1L26-29: Please split the long sentence. P1L37: place 'into account' immediately after 'take'.

---

## Referee Comment (RC2) · Anonymous Referee #2 · 27 Nov 2018

The paper addresses the importance of model evaluation and presents a robust method for evaluating the results from urban climate simulations. Overall, the paper is clear and well structured. The discussion on natural gap, observation bias and model bias is substantial, highlighting the problems existing in current modeling practices that the climatological modelers should pay more attention to. The study is valuable to be published in a high impact journal. I would suggest a minor revision in which the authors should focus more on evaluation framework and clarify some technical points.

Major Comments:

1. The focus of this paper should be the model evaluation. The authors may strengthen

the introduction and discussion of the evaluation framework in the following aspects:

(1) Presentation of the evaluation framework: the authors should summarize and present the evaluation framework in a visualized and more straightforward way (for example, using a workflow diagram).

(2) Justification for the evaluation tools: the authors should introduce more PSS theory and explain why it is suitable to evaluate the model for urban climate simulations. The same as the PDF analysis and other evaluation tools.

(3) Interpretation of the evaluation results: the authors kept using "acceptable" to describe the results. But how to define "acceptable"? What is the value of PSS would be considered as "not acceptable"? To make it a complete framework, the authors should provide guidelines to evaluate the results from the model evaluation.

(4) Intervals in PDF analysis: the authors use intervals of [-1, 1], [-2, 2], [-3, 3] for all variables in the PDF analysis. However, the significance of $\pm3$ degree in temperature change should have higher impact than $\pm3$ millimeter in precipitation. The authors should consider how to choose reasonable intervals for different variables.

(5) Selection of variables: the authors should state the rationale for choosing variables for model evaluation in your case study.

(6) Next steps: the authors should discuss the drawbacks of the proposed evaluation framework and provide suggestions for future research. It would be a plus if the authors provide the source codes and original datasets using in the model evaluation.

2. Although the inputs and setups in the modeling are critical to the model results, however, they are not the emphasis for this paper, and thus the modeling details should be listed in the appendix. On the other hand, a table of summarizing the evaluation results should be presented.

3. Here are some suggestions the authors may take into consideration for their future research by applying their proposed evaluation method in investigating the model

components and setups.

(1) New developed urban data: the authors developed four new sets of high-resolution urban data for modeling urban climate. What impact they have on the model results? Do they improve the overall performance of the model? If so, how much the improvement?

(2) Schemes of physics components: How to choose the schemes for each component? Would the selection of schemes have impacts on PSS scores?

Please also note the supplement to this comment:
https://www.geosci-model-dev-discuss.net/gmd-2018-220/gmd-2018-220-RC2-supplement.pdf

---

## Author Comment (AC2) · 28 Nov 2018

We appreciate reviewer 1 for spending time to review our paper and providing some valuable comments. We provided a quick response first and will provide a detail response later. The article is in pertinent response to the increasing presence of ambiguous or careless modelling practices in urban-scale climatology. Moreover, it is intended to state the necessity of model evaluation of urban-scale climatology modelling, draw attention within the community of urban climate modellers, and be a kick-off in reducing these window-dressing-like modelling practices. The purpose of this paper is for reminding the modellers the necessary of model evaluation in the urban climate mod-

elling practices rather than helping the model developer to improve the model. The modeller should conduct a systematic model evaluation to establish the trustworthiness of the new findings from an urban climate modelling because the model cannot be verified or validated. Moreover, we reminded that the modeller should be cautious to conclude a quantitative finding because it is impossible to identify the natural gap, observation bias, and model bias in the difference between observations and its corresponding modelled results. We are confident that this paper is important to climate modeller community because it point out the pain points which the existence of uncertainties of model affect the trustworthiness of the new findings and it is impossible to identify the uncertainties of model completely.

---

## Author Comment (AC3) · 28 Nov 2018

The reviewer made constructive comments to improve the presentation and structure of the paper. This is only a brief response and we will provide a detail response later. We plan to better organize the presentation of the proposed evaluation framework by including a clear workflow of the evaluation framework, more justification of PSS theory and other tools, and a summary table of evaluation results in our case study. With respect to the definition of 'acceptable', we will discuss it and summarized a framework of the practical grading guidelines, which shall be further refined given the proposed evaluation tools being applied in many other case studies. Moreover, we will also

reconsider the next steps and appendix. Furthermore, thanks for your interest in the developed high-resolution urban surface data. We are preparing another paper on it in which we compared the modeling results using the coarse urban land surface data provided by the WRF ARW model and the newly developed high-resolution urban land surface data. The paper should come out soon. Regarding the selection of schemes for the physics components, we will provide more details in the later version.

———————————————

---

## Author Comment (AC4) · 24 Dec 2018

Response to Reviewer 1

[Cover Letter] Dear Reviewer, We appreciate you for spending time to review our paper and providing some valuable comments. It is your valuable and insightful comments that led to possible improvements in the current version. The authors have carefully considered the comments and tried our best efforts to address every one of them. However, some revisions may still cannot meet your high standards. The authors welcome further constructive comments if any. We provided the point-by-point response first and will provide the updated version of the paper after proofreading complete.

[Figure]

Sincerely, Bo Huang, PhD bohuang@cuhk.edu.hk Professor, Department of Geography and Resource Management The Chinese University of Hong Kong

[General Comment] This study evaluates performance of the WRF model in terms of high-resolution urban climate modelling over an area encompassing two big cities, Shenzhen and Hong Kong. The chosen area of Shenzhen is heavily urbanized but only a small part of Hong Kong is urbanized. Perkins skill score is used as a major evaluation method throughout the evaluation. The authors argue that their study has proposed a methodological framework for evaluating model performance in high resolution urban climate simulation. I think this work is useful and has provided some information about high-resolution urban climate modelling applied to south China. I very much appreciate the authors' efforts to pursue this kind of modelling work. However, I feel that the manuscript in the current form cannot be accepted for publication. At a minimum, I would suggest some necessary revisions to make the paper publishable in the journal. But to engender a stronger paper, I feel that more extensive work might have to be done. I will leave it to the editor to decide whether such extensive work is required.

Response: The article is in pertinent response to the increasing presence of ambiguous or careless modelling practices in urban-scale climatology. It intended to state the necessity of model evaluation of urban-scale climatology modelling, draw attention within the community of urban climate modellers, and be a kick-off in reducing these window-dressing-like modelling practices. Therefore, the purpose of this paper is for reminding the modellers the necessary of model evaluation in the urban climate modelling practices rather than helping the model developer to improve the model. Moreover, the modeller should conduct a systematic model evaluation to establish the trustworthiness of the new findings from an urban climate modelling because the model cannot be verified or validated. Furthermore, we reminded that the modeller should be cautious to conclude a quantitative findings because it is impossible to identify the natural gap, observation bias, and model bias in the difference between observations and

its corresponding modelled results. To sum up, we are confident that this paper is important to urban climate modeller community because it points out the pain points which the existence of uncertainties of model affect the trustworthiness of the new findings and it is impossible to identify the uncertainties of model completely.

[Major Comment 1] The introduction should be reformulated with greater care. The authors should survey the literature more thoroughly. Only a few papers are mentioned in the introductory section. I suggest the authors give a good overview of the existing studies on the topic and point out the limitations of the past studies and challenges/constrains. Identifying a gap or proposing a new method as well as outlining the contributions of the study is also helpful. Response: We added some new related literatures in Section 1 to emphasize the importance of model evaluation in urban climate modelling, the fact that the modellers paid the less attention in their modelling practices and the values of this paper.

[Major Comment 2] The data and methodology section should be structured in a more logical way. I think the authors could place model description and experiment/model setup before evaluation method. Overall, both section 2 and 3 are a bit confusing. The introduction of the model is lacking. The authors should clearly articulate what has been done and how it has been done. This can aid the readers in understanding the experiment setup/design.

Response: We revised Section 2 for improving clarity and provided more information about model description and setup in Supplementary Material. Moreover, we will submit another paper for describing all details about our urban climate modelling practice which includes the suggestions for modelling process, the atmospheric model design, model setup, primary data processing method, and a quality assurance framework.

[Major Comment 3] In section 2.1, more details about the new dataset developed by the authors should be offered. Response: We provided more details about the land surface dataset in Supplementary Material.

The reasons for focusing on the simulations in the year of 2010 should be discussed.

Response: The reason is the data limitation that we only have the land surface data and observation data in 2010. We mentioned it in the revision of the paper.

In section 2.2, more details should be provided as to the four-day segment simulations.

Response: We provided more details about four-day segment in Supplementary Material.

Did the model read in restart files every four days to continue the simulation?

Response: No. Each four-days simulation segment is a separated simulation.

How may a different simulation strategy affect the modelling results?

Response: The different simulation strategies relate with the different spin-up method, which affect the modelling results. We added a small discussion about it in Section 2.2.

In section 2.3, instead of just giving two tables, I think more detailed descriptions of the data should be given. How are the comparisons between model output (grid points) and observations (stations) made? Representativeness of the observations and potential biases should be discussed. The authors should also indicate the reasons for choosing evaluation variables.

Response: We added more details about the modelling variables and observation data in Section 2.3.

[Major Comment 4] In section 2.4, no references are cited regarding the Perkins skill score. Is this a suitable method for this study? There should at least be some discussion. Authors should also discuss whether this method is suitable for all the variables evaluated in the study.

Response: We conducted a small discussion about the evaluation tools in Section 2.4.

[Major Comment 5] In section 3, choosing of the parameterization schemes needs

discussion.

Response: We conducted a small discussion of the parameterization schemes in Supplementary Material.

[Major Comment 6] I think the authors should tune down many of their arguments throughout the paper to avoid overstating (e.g., P2L25-26). For example, I don't see any strong methodological framework being discussed and described in the text.

Response: We enhanced the description of methodological framework for supporting our statement. We add a subsection (2.3 A Methodological Framework for Urban Climate Model Evaluation) to describe more details about the methodological framework.

[Major Comment 7] I have the impression that the authors have been too obsessed with 'good results' when evaluating the model's performance. Discussing 'good results' and 'bad results' at the same time, in my opinion, is fair. It's perhaps more important to identify areas for improvements.

Response: This paper intended to state the necessity of model evaluation of urban-scale climatology modelling and provided a methodological framework of model evaluation to help the modeller to establish the trustworthiness of modelling results, and accordingly it focused on the modelling performance rather than the identifying areas for improvements in order to help the model developer improving the model. We added an explanation in Section 1 to emphasize the focus of this paper.

[Major Comment 8] The structure and writing are too repetitive in section 4. This is also true for the figures. The number of figures may be reduced.

Response: We did our best to rewrite Section 4. Moreover, we moved some figures to Supplementary Material for reducing the number of figures in the paper.

While the focus of the paper as stated in the paper is on the urban climate simulation, evaluation seems to be applied to also the vast rural regions. The authors should clarify this.

Response: Yes. The methodological framework of model evaluation also can be applied in the local scale climate simulation wherever in urban or non-urban areas. We added an explanation in Section 5.

I suggest the authors focus on the most important aspects of the urban climate simulation. I would suggest some points (see following) for the authors to consider and they should further develop a better evaluation framework.

Response: Thank you very much for your suggestions. We added a subsection (2.3 A Methodological Framework for Urban Climate Model Evaluation) to describe more details about the methodological framework, which included a graphical presentation of the framework, the grading guidelines for PSS and PDF of the difference, and theoretical explanation to the statistic tools applied in model evaluation.

-Some basic ability of the model such as spatial distribution temperature/precipitation and diurnal cycles of temperature must be assessed.

Response: The diurnal cycles of 2m air temperature had be assessed in Subsection 4.1. We provided more comparisons of meteorological variables between urban and nonurban areas in Supplementary Material.

- The weather and climate variability in the study area is strongly associated with the monsoon flow. So the investigation of the simulation of precipitation and temperature is rather important. Both the spatial distribution (not found in any of the figures in the paper) and temporal variability should be considered.

Response: We agreed that the climate variability in the study area is strongly associated with monsoon flow. However, the monsoon flow is a mesoscale meteorological behaviour, and accordingly, it is not associated the spatial distribution of precipitation and temperature in a local scale region. The spatial distribution of precipitation and temperature is strongly associated with the local land surface attributes. Therefore, we added some discussions in Section 4 about the relationship in the spatial distribu-

tion between 2m air temperature and land surface, also the relationship in the spatial distribution between precipitation and land surface. Moreover, we agreed that the seasonal variations in temperature and precipitation are associated with monsoon flow, especially precipitation. Therefore, we added some discussions in Section 4 about the relationship between the monsoon flow and the seasonal variation of precipitation, also between the monsoon flow and the seasonal variation of 2m air temperature.

In particular, the authors may identify some strong urbanization impacts on the precipitation (e.g., precipitation maxima) and temperature (e.g., urban heat island). The model's ability to capture these effects is essential.

Response: It needed the observation data before and after urbanization to evaluate the urbanization impacts on the precipitation and temperature. We cannot provide these evaluations because we don't have these observation data. However, we added some discussions in Section 4 about the relationship in the spatial distribution between 2m air temperature and land surface, also the relationship in the spatial distribution between precipitation and land surface.

In addition, simulation of sea breeze, wind distribution, boundary layer variability, and stability of the atmosphere should be examined.

Response: We agreed that the land-sea breeze exists in the coastal city, and accordingly we provided a discussion about the modelled land-sea breeze in Section 4. These modelled meteorological features (boundary layer variability and atmospheric stability) cannot be examined by the observation due to the unavailability of its corresponding observation data. It is meaningless that the modelled meteorological features examined without comparison with the observation. Therefore, we didn't provide the examination on the these two meteorological features.

The impact of urbanization on the air quality may also be discussed.

Response: This study focused on providing a methodological framework of urban cli-
mate model evaluation. The impact of urbanization on the air quality is another big topic which is out of the research scope in this study. Therefore, we didn't provide an discussion on it in this paper.

- The evaluation can be done separately for different seasons. The evaluation should focus on the most important aspects of urban climate/weather.

Response: Actually, the figures included the information of the monthly variations. We provided more details about the seasonal variations of the evaluation in Section 4.

- The scientific value can be enhanced if the authors can demonstrate how the model behaves in simulating the extreme precipitation events or heat wave/cold surge events, and How and to what extent these events may be related to the urbanization.

Response: Thank you very much for your suggestions. However, our study focused on reminding the urban climate modeller the importance of model evaluation in establishing the trustworthiness of modelling results and provided a methodological framework of model evaluation, and accordingly we didn't put too much effort on model performance on simulating the extreme events. In the revision, we added some discussions about the performance on simulation the extreme events on Sections 4 and 5.

- The model's performance between different regions in the study area and between rural and urban regions can also be compared.

Response: Thank you very much for your suggestions. We added more figures about the model's performance in urban and non-urban areas in Supplementary Material.

[Major Comment 9] The figures can be better designed and drawn. Captions of the figures should provide more information. The language could also be improved.

Response: Thank you very much for your suggestions. We did our best to improve the language and the captions of the figures.

[Minor Comment] Minor comments: The authors should check carefully the use of

words and sentences throughout the paper. I suggest some serious edits/revisions. I list only some of the examples. P1L15: add 'have' before paid. P1L26-29: Please split the long sentence. P1L37: place 'into account' immediately after 'take'.

Response: Thank you very much for your suggestions. We did our best to check the paper, corrected the language errors and rewrote the long sentences for improving the readability.

Please also note the supplement to this comment:
https://www.geosci-model-dev-discuss.net/gmd-2018-220/gmd-2018-220-AC4-supplement.pdf

───────────────────────────────

---

## Author Comment (AC5) · 24 Dec 2018

Response to Reviewer 2

[Cover Letter] Dear Reviewer, We appreciate you for spending time to review our paper and providing some valuable comments. It is your valuable and insightful comments that led to possible improvements in the current version. The authors have carefully considered the comments and tried our best efforts to address every one of them. However, some revisions may still cannot meet your high standards. The authors welcome further constructive comments if any. We provided the point-by-point response first and will provide the updated version of the paper after proofreading complete.

[Figure]

Sincerely,

Bo Huang, PhD

bohuang@cuhk.edu.hk

Professor, Department of Geography and Resource Management

The Chinese University of Hong Kong

[General Comment] The paper addresses the importance of model evaluation and presents a robust method for evaluating the results from urban climate simulations. Overall, the paper is clear and well structured. The discussion on natural gap, observation bias and model bias is substantial, highlighting the problems existing in current modeling practices that the climatological modelers should pay more attention to. The study is valuable to be published in a high impact journal. I would suggest a minor revision in which the authors should focus more on evaluation framework and clarify some technical points. Response: The reviewer made constructive comments to improve the presentation and structure of the paper. We better organized the presentation of the proposed evaluation framework by including a clear workflow of the evaluation framework, more justification of PSS theory and other tools, and a summary table of evaluation results in our case study. With respect to the definition of 'acceptable', we discussed it and summarized a framework of the practical grading guidelines, which shall be further refined given the proposed evaluation tools being applied in many other case studies. Moreover, thanks for your interest in the developed high-resolution urban surface data. We are preparing another paper on it in which we compared the modeling results using the coarse urban land surface data provided by the WRF ARW model and the newly developed high-resolution urban land surface data. The paper should come out soon. Furthermore, regarding the selection of schemes for the physics components, provided more details in the later version.

Major Comments: [Comment 1] The focus of this paper should be the model evaluation. The authors may strengthen the introduction and discussion of the evaluation framework in the following aspects:

(1) Presentation of the evaluation framework: the authors should summarize and present the evaluation framework in a visualized and more straightforward way (for example, using a workflow diagram). Response: Thank you for your suggestions. We added a detailed explanaion of the proposed model evaluation framework in Section 2.2. We also added a workflow diagram for better presentation of the proposed model evaluation framework. We hope the extended explanations made the proposed framwork easier to understand.

(2) Justification for the evaluation tools: the authors should introduce more PSS theory and explain why it is suitable to evaluate the model for urban climate simulations. The same as the PDF analysis and other evaluation tools. Response: The importance of examining climate statistics other than climate means is not new (Katz and Brown 1992; Boer and Lambert 2001). The descriptive statistics are useful in providing aggregated information on the distribution of the attributes, but they can be very misleading since very different distributions can lead to similar descriptive statistics, and these aggregated metrics can be sensitive to outliers. Therefore, we examine not only the descriptive statistics but also metrics regarding the statistical distributions of modeled and observed meteorological attributes. The advantages of PDF and PSS for climate statistics have been discussed by Perkins et al. (2007). We have revised accordingly in Section 2.2.

(3) Interpretation of the evaluation results: the authors kept using "acceptable" to describe the results. But how to define "acceptable"? What is the value of PSS would be considered as "not acceptable"? To make it a complete framework, the authors should provide guidelines to evaluate the results from the model evaluation. Response: Thanks for your great suggestions. A reasonable definition of "acceptable" would improve the novelty of this manuscript. We summerized the 72 monthly analysis for 6 meteorological attributes in our case study. The PSS values generally followed a normal distribution ranging from 0.444 to 0.886 with an average of 0.660 and a standard deviation of 0.098. Therefore, the PSS values are larger than 0.500 with a probability of 95%. Based on our results, we define 2 criteria for "acceptable" high-resolution urban climate simulations: 1) Yearly average PSS $\geq$ 0.550; 2) For each meteorological attribute, PSS $\geq$ 0.500 with a confidence interval of 0.500. Compared to the case studies in Perkins et al. (2007), the lower bounds for PSS in our standard was lower, which is due to the increased resolution in our simulations. We are fully aware that, despite the sophisticated analysis we have conducted on different spatial and temporal scales, it is difficult to define 'acceptable' using results from one case study. In this vein, the proposed standard of 'acceptable' for high-resolution urban climate simulations based on our case study was meant to be the starting point and to be improved by future case studies using the proposed model evaluation framework.

(4) Intervals in PDF analysis: the authors use intervals of [-1, 1], [-2, 2], [-3, 3] for all variables in the PDF analysis. However, the significance of 3 degree in temperature change should have higher impact than 3 millimeter in precipitation. The authors should consider how to choose reasonable intervals for different variables. Response: Thank you very much for your comment. Indeed, the same intervals have very different meanings for different meteorological attributes. Therefore, we are planning to use the standard deviations as intervals of the PDF analysis instead of fixed intervals.

(5) Selection of variables: the authors should state the rationale for choosing variables for model evaluation in your case study. Response: We included all meteorological variables that are meaningful for urban climate analysis in the model evalutaion.

(6) Next steps: the authors should discuss the drawbacks of the proposed evaluation framework and provide suggestions for future research. It would be a plus if the authors provide the source codes and original datasets using in the model evaluation. Response: Thank you for your comment. We added more discussions on drawback of this study and possible future directions in the Conclusions. We are planning to open the entire dataset in another paper specifically on the development of the high-resolution

urban land surface data under review. "This study is not perfect. The effects of the selected physical components on the evaluated modelling accuracy is not clear, which requires further control experiments. Also, the effects of the refined urban land surface datasets on the evaluated modelling accuracy also requires further discussions."

[Comment 2] Although the inputs and setups in the modeling are critical to the model results, however, they are not the emphasis for this paper, and thus the modeling details should be listed in the appendix. On the other hand, a table of summarizing the evaluation results should be presented. Response: Thank you for your advise. We are including Section 3 in Supplementary Material.

[Comment 3] Here are some suggestions the authors may take into consideration for their future research by applying their proposed evaluation method in investigating the model components and setups. Response: Thank you very much for your suggestions. We definitely agree with you that many more can be done and we will continue to thrive in this direction.

(1) New developed urban data: the authors developed four new sets of high-resolution urban data for modeling urban climate. What impact they have on the model results? Do they improve the overall performance of the model? If so, how much the improvement? Response: Thank you for your comment. That is a great point for future research. Acturally, we have another manuscript under review focused on the development of the high-resolution urban land surface dataset and its effects on the reliability of climate simulation. Please refer to Archive. We can briefly introduce results from the other paper. People would naturally expect more accuracte modeling results with more accuracte input data. Unexpectedly, we find that high-resolution urban land surface datasets could either increase or decrease the evaluated reliability of simulation results, which is probably why not all modelers refine the land surface input data before the simulation. We believe the reason for this phenomenon is due to inperfect model and inperfect model evaluation methods. First, imperfectness in the detailed physical processes included in the model. This is the root why more accurate input data

does not necessarily lead to more accurate simulation results. Second, it's inperfect to compare the grid-based simulation results with point-based observations. Moreover, the evaluated reliability cannot be compared across scales since high-resolution simulation contain many more details and will naturally decrease the evaluated reliability. Nevertheless, even the decrease in model evaluation metrics does not mean that the simulation results are less accuracte. We argue that providing more accurate input data into the model is the only way to prevent the 'garbage in, garbage out' effect, motivate us to refine the model itself and model evaluation methods, and lead us towards better modeling practice.

(2) Schemes of physics components: How to choose the schemes for each component? Would the selection of schemes have impacts on PSS scores? Response: That is another great point for future research. The interactions among selected physical components are very complex and so it can be difficult to provide solid foundations for the selections. One way out is to compare the accuracy of simulation results using different combinations of physical components. However, there are three possible challenges: 1) many variables are involved in this process, including physical components, model parameters, and spatial-temporal resolutions, which makes a very large solution space. It would be compotationally expensive to try every combination; 2) Evidence from a certain study area and time period may not be transferable to other study areas or time periods; 3) proper model evaluation metric. Comparing grid-based modeling results with point-based simulations are naturally biased. Better evaluaed accuracy does not necessarily mean better quality of simulation results. Therefore, we think some evidence can definitely be provided from experiments using specific study area, time period, model parameter, and spatial-temporal resolution. But it would be difficult to provide more general insights for the seleciton of physical components. Theoretical discussions and the 'try and error' process are still vital in refining such selections.

Please also note the supplement to this comment:

https://www.geosci-model-dev-discuss.net/gmd-2018-220/gmd-2018-220-AC5-supplement.pdf

[Figure]

[Figure]

*Figure 1. Histogram of the Perkins skill score for 72 monthly PDF analysis. A normal distribution was fit. The red dashed line indicate the lower bound for 95% confidence (PSS=0.500).*

**Fig. 1.**

---

## Author Comment (AC6) · 22 Jan 2019

Response to Reviewer 1 [Cover Letter] Dear Reviewer, We appreciate your precious time in reviewing our paper and valuable comments. It is your valuable and insightful comments that led to possible improvements in the current version. The authors have carefully considered the comments and tried our best to address every one of them. We hope the revisions meet your high standards. The authors welcome further constructive comments if any. We provided the point-by-point response as below. Modifications in the manuscript are highlighted in red.

Sincerely, Bo Huang, PhD bohuang@cuhk.edu.hk Professor, Department of Geogra-

phy and Resource Management The Chinese University of Hong Kong

[General Comment] This study evaluates performance of the WRF model in terms of high-resolution urban climate modelling over an area encompassing two big cities, Shenzhen and Hong Kong. The chosen area of Shenzhen is heavily urbanized but only a small part of Hong Kong is urbanized. Perkins skill score is used as a major evaluation method throughout the evaluation. The authors argue that their study has proposed a methodological framework for evaluating model performance in high resolution urban climate simulation. I think this work is useful and has provided some information about high-resolution urban climate modelling applied to south China. I very much appreciate the authors' efforts to pursue this kind of modelling work. However, I feel that the manuscript in the current form cannot be accepted for publication. At a minimum, I would suggest some necessary revisions to make the paper publishable in the journal. But to engender a stronger paper, I feel that more extensive work might have to be done. I will leave it to the editor to decide whether such extensive work is required.

Response: The article is in pertinent response to the increasing presence of ambiguous or careless modelling practices in urban-scale climatology. It intended to state the necessity of model evaluation in urban-scale climatology modelling, draw attention within the community of urban climate modellers, and be a kick-off in reducing these window-dressing-like modelling practices. Therefore, the purpose of this paper is to remind modellers of the necessity of model evaluation in the urban climate modelling practices rather than helping to improve the model. Moreover, the modeller should conduct a systematic model evaluation to establish the trustworthiness of the new findings from an urban climate modelling since the model cannot be verified or validated. Furthermore, we reminded that the modeller should be cautious to conclude quantitative conclusions because it is impossible to differentiate the natural gap, observation bias, and model bias in the difference between observations and its corresponding modelled results. To sum up, we are confident that this paper is important to the urban climate

modeller community as it points out the pain points, that is, model uncertainties affect the trustworthiness of the new findings and it is impossible to identify the uncertainties of model completely.

[Major Comment 1] The introduction should be reformulated with greater care. The authors should survey the literature more thoroughly. Only a few papers are mentioned in the introductory section. I suggest the authors give a good overview of the existing studies on the topic and point out the limitations of the past studies and challenges/ constrains. Identifying a gap or proposing a new method as well as outlining the contributions of the study is also helpful.

Response: We added some new related literatures in Section 1 to emphasize the importance of model evaluation in urban climate modelling and the fact that modellers paid minimal attention in their modelling practices. Moreover, we identified the systematic framework for model evaluation as the research gap in the urban climate modelling community and outlined the values of this paper in Section 1. (Pg2, Ln27-34)

[Major Comment 2] The data and methodology section should be structured in a more logical way. I think the authors could place model description and experiment/model setup before evaluation method. Overall, both section 2 and 3 are a bit confusing. The introduction of the model is lacking. The authors should clearly articulate what has been done and how it has been done. This can aid the readers in understanding the experiment setup/design.

Response: We revised Section 2 to improve clarity and provided more information about model description in Section 2.1 [Pg3, Ln2-23] and set-up in Section S4 of Supplementary Material. Moreover, we will submit another paper to describe all details about the high-resolution urban climate modelling including suggestions for modelling process, the design of the atmospheric model, model set-up, primary data processing, and a framework for quality assurance.

[Major Comment 3] In section 2.1, more details about the new dataset developed by

the authors should be offered.

Response: We provided more details about the developed land surface dataset in Section S2 of Supplementary Material. Moreover, we will submit another paper to provide all details about this urban land surface dataset later.

The reasons for focusing on the simulations in the year of 2010 should be discussed.

Response: We selected the year of 2010 since it was the latest year that a complete government-initiated land survey was conducted, which provided access to high-quality field-surveyed land cover data that is crucial for the climate simulation. It requires various data sources for the development of the new land surface dataset, high-resolution urban climate simulation and model evaluation, and we have datasets available around the year of 2010. We mentioned it in the revision of the paper [Pg3, Ln11-12].

In section 2.2, more details should be provided as to the four-day segment simulations.

Response: We provided more details about four-day segments in Section S3 of Supplementary Material.

Did the model read in restart files every four days to continue the simulation?

Response: No. Each four-days simulation segment is a separated simulation.

How may a different simulation strategy affect the modelling results?

Response: Different simulation strategies is associated with the different spin-up method, which affect the modelling results. We added a small discussion about it in Section S3 of the Supplementary Material.

In section 2.3, instead of just giving two tables, I think more detailed descriptions of the data should be given. How are the comparisons between model output (grid points) and observations (stations) made?

Response: We already discussed these comparisons in Section 4. Moreover, we

added some details about the comparison in Subsection 2.3 [Pg4, Ln14-19]. Furthermore, we would like to provide the source codes of the evaluation software packages to the readers for easy replication.

Representativeness of the observations and potential biases should be discussed.

Response: We added more details about the observation datasets in Section S5 of the Supplementary Material.

The authors should also indicate the reasons for choosing evaluation variables.

Response: We added reasons for choosing evaluation variables in Subsection 2.3 [Pg4, Ln9-12].

[Major Comment 4] In section 2.4, no references are cited regarding the Perkins skill score. Is this a suitable method for this study? There should at least be some discussion. Authors should also discuss whether this method is suitable for all the variables evaluated in the study.

Response: We conducted a small discussion about the evaluation tools in Subsection 2.2 [Pg3, Ln24 – Pg4, Ln4].

[Major Comment 5] In section 3, choosing of the parameterization schemes needs discussion.

Response: We conducted a small discussion of the selection of parameterization schemes in Section S4 of Supplementary Material.

[Major Comment 6] I think the authors should tune down many of their arguments throughout the paper to avoid overstating (e.g., P2L25-26). For example, I don't see any strong methodological framework being discussed and described in the text.

Response: We enhanced the description of methodological framework to support our statement. We add a subsection (2.2 A Methodological Framework for Urban Climate Model Evaluation) to include more details about the methodological framework [Pg3,

[Figure]

Ln24 – Pg4, Ln4].

[Major Comment 7] I have the impression that the authors have been too obsessed with 'good results' when evaluating the model's performance. Discussing 'good results' and 'bad results' at the same time, in my opinion, is fair. It's perhaps more important to identify areas for improvements.

Response: This manuscript intended to state the necessity of model evaluation of urban-scale climatology modelling and to provide a methodological framework of model evaluation to help modellers to establish the trustworthiness of modelling results, and accordingly it focused on the modelling performance rather than to help the model developers improving the model. We added an explanation in Section 1 to emphasize the focus of this paper [Pg2, Ln28-35].

[Major Comment 8] The structure and writing are too repetitive in section 4. This is also true for the figures. The number of figures may be reduced.

Response: We did our best to rewrite Section 4. Moreover, we moved some figures to Supplementary Material for reducing the number of figures in the paper.

While the focus of the paper as stated in the paper is on the urban climate simulation, evaluation seems to be applied to also the vast rural regions. The authors should clarify this.

Response: Yes. The methodological framework of model evaluation also can be applied in the local scale climate simulation wherever in urban or non-urban areas. We added an explanation in Section 5 [Pg11, Ln24-25].

I suggest the authors focus on the most important aspects of the urban climate simulation. I would suggest some points (see following) for the authors to consider and they should further develop a better evaluation framework.

Response: Thank you very much for your suggestions. We added a subsection (2.2 The Methodological Framework for Urban Climate Model Evaluation) to describe more

details about the methodological framework, which included a theoretical explanation to the statistic tools applied in model evaluation[Pg3, Ln24 – Pg4, Ln4]. Moreover, we added Subsection 2.4 , which included a graphical presentation of the workflow of model evaluation, the guideline for checking the descriptive statistics figures and the grading guidelines for PSS and PDF of the difference [Pg5, Ln2 – Pg6, Ln6].

-Some basic ability of the model such as spatial distribution temperature/precipitation and diurnal cycles of temperature must be assessed.

Response: The difference in the surface temperature between in urban and non-urban areas (spatial distribution of temperature) had be assessed in Figure 9. The difference of precipitation between in urban and non-urban areas is not significant. The diurnal cycles of 2-meters air temperature had be assessed in Figure 3.

- The weather and climate variability in the study area is strongly associated with the monsoon flow. So the investigation of the simulation of precipitation and temperature is rather important. Both the spatial distribution (not found in any of the figures in the paper) and temporal variability should be considered.

Response: We agreed that the climate variability in the study area is strongly associated with the monsoon flow. However, the monsoon flow is a mesoscale meteorological behaviour and so it is not associated with the spatial distribution of precipitation and temperature at the local scale. The spatial distribution of temperature is strongly associated with the local land surface attributes. Therefore, we added some discussions in Subsection 5.3 about the relationship in the spatial distribution between 2-m air temperature and land surface temperature [Pg11, Ln5 – 25]. Moreover, we agree that seasonal variations in temperature and precipitation are associated with monsoon flow, especially precipitation. Therefore, we added some discussions in Subsection 5.3 on the relationship between the monsoon flow and the seasonal variation of precipitation, and the relationship between the monsoon flow and the seasonal variation of 2-m air temperature [Pg11, Ln5 – 25].

In particular, the authors may identify some strong urbanization impacts on the precipitation (e.g., precipitation maxima) and temperature (e.g., urban heat island). The model's ability to capture these effects is essential.

Response: Observational data before and after the urbanization process are needed to evaluate the urbanization impacts on the precipitation and temperature. We cannot provide these evaluations because we don't have these observation data. However, we added some discussions in Section 5.3 on the relationship between the spatial distribution of the 2-m air temperature and the land surface temperature, and also the relationship between the spatial distribution of precipitation and land surface temperature [Pg11, Ln5 – 25].

In addition, simulation of sea breeze, wind distribution, boundary layer variability, and stability of the atmosphere should be examined.

Response: We agree that the land-sea breeze exists in the coastal city, and sowe provided a discussion about the modelled land-sea breeze in Subsection 5.3 [Pg11, Ln5 – 25]. These modelled meteorological features (boundary layer variability and atmospheric stability) cannot be examined by the observation due to the unavailability of corresponding observation data. Examining modelled meteorological features is meaningless without comparison with observations. Therefore, we didn't provide the examination of these two meteorological features.

The impact of urbanization on the air quality may also be discussed.

Response: This study focused on providing a methodological framework for the evaluation of urban climate models. The impact of urbanization on air quality is another big topic beyond the research scope of this study.

- The evaluation can be done separately for different seasons. The evaluation should focus on the most important aspects of urban climate/weather.

Response: Actually, all figures includes the information of monthly variations in this

paper, while some also show the seasonal variations. Moreover, we emphasize that the model evaluation should focus on the comparison between the modelled variables with its corresponding observed ones. Furthermore, we added a small discussion about it in Subsection 5.3 [Pg11, Ln5 – 25].

- The scientific value can be enhanced if the authors can demonstrate how the model behaves in simulating the extreme precipitation events or heat wave/cold surge events, and How and to what extent these events may be related to the urbanization.

Response: Thank you very much for your suggestions. However, our study focused on reminding the urban climate modeller of the importance of model evaluation and establishing the trustworthiness of modelling results. We also provided a methodological framework of model evaluation, and so we didn't put too much effort on the modelling performance of simulating the extreme events. In this revision, we added some discussions about the capabilities on the simulations of the extreme events on Sections 5.3 [Pg11, Ln5 – 25].

- The model's performance between different regions in the study area and between rural and urban regions can also be compared.

Response: Thank you very much for your suggestions. We added more figures on the model's performance in urban and non-urban areas in Section S6 of Supplementary Material.

[Major Comment 9] The figures can be better designed and drawn. Captions of the figures should provide more information. The language could also be improved.

Response: Thank you very much for your suggestions. We did our best to improve the language and the figure captions.

[Minor Comment] Minor comments: The authors should check carefully the use of words and sentences throughout the paper. I suggest some serious edits/revisions. I list only some of the examples. P1L15: add 'have' before paid. P1L26-29: Please split

the long sentence. P1L37: place 'into account' immediately after 'take'.

Response: Thank you very much for your suggestions. We did our best to check the paper, corrected the language errors and rewrote the long sentences to impro

Please also note the supplement to this comment: https://www.geosci-model-dev-discuss.net/gmd-2018-220/gmd-2018-220-AC6-supplement.pdf

[Figure]

**Supplement:**

[revised manuscript text omitted]
 wasn't provided in the previous literatures. It is especially a research gap in urban climate modelling community to proposing a systematic framework and methods for model evaluation. Thus, in this paper, we dig deeply into the model evaluation and propose a systematic framework and methods for evaluating model results from multiple perspectives, to benefit future studies with more choices for model quality control and make urban scale simulation more robust. Moreover, we also provide a case analysis of the interval between the modelled atmospheric variable and its corresponding observed one.

The remainder of this paper is organized as follows. Section 2 introduces the proposed framework for model evaluation, experimental design, and data used for modelling and model evaluation. Section 3 introduces the technical preparation for the urban climate simulation. Section 4 presents various results of the proposed model evaluation methods in our case study. Section 5 concludes the paper with discussions.

**2 Methodology**

**2.1 Urban Climate Modelling**

[revised manuscript text omitted]

5    **Table 1. An Evaluation Framework for Urban Climate Modelling**

| Metrics | | Temporal Perspectives | | |
|---|---|---|---|---|
| **Statistical Perspectives** | **Method** | **Annual** | **Monthly** | **Daily** |
| Descriptive Statistics | Temporal comparison | Annual Pattern | Monthly Pattern | Diurnal pattern |
| Statistical Distributions | Overlap of the probability density functions of modelled and observed data | Annual mean PSS | Monthly PSS | |
| | Probability density function of the difference between modelled and observed data | Annual Mean score of the specified interval | Distribution and the score of the specified intervals | |

**2.3 Observation Datasets and Modelled Variables for Model Evaluation**

In existing literatures, Numerical Weather Prediction (NWP) models are typically evaluated by comparing the spatial-temporal patterns of the modelled variables with those of its corresponding near-surface observations. Moreover, four prognostic variables (2-meter air temperature, surface temperature, 10-meter wind at u direction and 10-meters wind at v
10    direction) and three diagnostic variables (accumulated total cumulus precipitation, accumulated total grid  precipitation and 2-meter relative humidity) were chosen as the modelled variables for the comparison because these variables are the critical variables in the prognostic and diagnostic equations in the NWP model.

Table 2 shows the observation datasets we used for the comparison between modelled results and observations in the inner-most domain. Table 3 lists the modelled variables and its corresponding observed ones in the model evaluation. The
15    observation datasets are the point data except the MODIS dataset which is the grid data. All the modelled variables are grid data. The comparisons between modelled variables with its corresponding observed ones are comparisons  between the grid value of the modelled variable with its point value matched by to geographical locations,  except for the comparison between the modelled surface temperature and its corresponding observation retrieved from MODIS imagery.

**Table 2: Observation Datasets.**

| Observation Datasets | Sources |
|---|---|
| 2010 PRD 2-Meters Air Temperature | Meteorological Bureau of Shenzhen Municipality |
| 2010 PRD 10-Meters Wind Speed | |
| 2010 PRD Precipitation | |
| 2010 PRD Relative Humidity | |
| 2010 MODIS/Aqua Land Surface Temperature and Emissivity (LST/E) product | NASA EOSDIS Land Processes DAAC, USGS Earth Resources Observation and Science (EROS) Center |

**Table 3: Modelled Variables for Model Evaluation.**

| Modelled Variables for Model Evaluation | | Corresponding Observation Datasets |
|---|---|---|
| **Name** | **Description** | |
| T2 | 2-meters air temperature | 2010 PRD 2-Meters Air Temperature |
| TSK | Surface temperature | 2010 MODIS/Aqua Land Surface Temperature and Emissivity (LST/E) product |
| U10 | 10-meters wind at U direction | 2010 PRD 10-Meters Wind Speed |
| V10 | 10-meters wind at V direction | |
| RAINC | Accumulated total cumulus precipitation | 2010 PRD Precipitation |
| RAINNC | Accumulated total grid scale precipitation | |
| RH2 | 2-meters relative humidity | 2010 PRD Relative Humidity |

**2.4 Workflow for Model Evaluation**

Following the proposed framework, we used four statistic figures (Table 4) and designed a workflow (Figure 1) in the practice of model evaluation. We first conducted a data processing operation (run by a program) on each pair of the raw observation dataset and its corresponding modelled result to produce an evaluation 3D-matrix which consists of a one-year temporal series of 2D-matrixes. Each non-empty element of the 2D-matrixes geographically corresponds with a pair of observation data and its corresponding modelled one a time point. A location map of meteorological observations was also produced if the raw observation dataset is a meteorological observation dataset. Secondly, we conducted the figure plotting operations (run by the programs) on each evaluation 3D-matrix step by step to produce a series of statistic figures. Specially, we designed a guideline (Table 5) for specifying the intervals in the PDFD figure, which are used for measuring the accuracy. Finally, we conducted grading or checking operations on these statistic figures artificially base on the guidelines for grading (Table 6) or checking (Table 7).

**Table 4: The Statistic Figures for Model Evaluation.**

| Statistical Perspectives | Methods | |
|---|---|---|
| | **Statistic Figures** | **Usage** |
| **Descriptive Statistics** | Temporal Comparison of Spatial Variation (TCSV) | It is used to temporally compares two variables' spatial variation ranges and median in the whole year. |
| | Diurnal Variation (DV) | It is used to temporally compares the diurnal variations of two variables' spatial variation ranges and median in the whole year. |
| **Statistical Distributions** | Probability Density Function of Difference (PDFD) | It is used to show the probability density of difference between the modeled variable and its corresponding observed one. |
| | Perkins Skill Score (PSS) | It is used for revealing quantifiably the extent of overlap between the observed and modeled variables' Probability Density Function (PDF). A value of 1 indicates a perfect modeling of the observation. On the contrary, a value of 0 means the worst simulation. |

**Table 5: The Guideline for Specifying the Interval in a PDFD Figure.**

| No. | The Range of Coefficient (A) | Intervals |
|-----|------------------------------|-----------|
| 1 | 0.1 – 0.2 | |
| 2 | 0.2 – 0.4 | [-Aσ, Aσ] |
| 3 | 0.4 – 0.6 | |

Remark: σ is the annual mean value of the monthly standard deviation of the modelled variable

Table 6: The Guideline for Grading.

| Range | | Grading |
|-------|--|---------|
| PSS | Accuracy(a) | |
| 0.7 ≤ PSS ≤ 1 | 70% ≤ a ≤ 100% | Good |
| 0.7 > PSS ≥ 0.5 | 70% > a ≥ 50 % | Acceptable |
| PSS < 0.5 | a < 50% | Unacceptable |

Remark: Accuracy is the PDFD value of interval 2 or interval 3.

Table 7: The Guideline for Checking.

[revised manuscript text omitted]

**5.3 The Capabilities of WRF ARW/Noah LSM/SLUCM in Simulating Meteorological Phenomena**

This paper aims to present a standardized methodological framework that can be used in all regions to evaluate the modelling performance rather than to examine the capabilities of the model in simulating meteorological phenomena in a specified region. Therefore, we proposed model evaluation methods for the comparison between modelled variables and its corresponding observed ones rather than investigating the meteorological phenomena in the modelled results. However, the capabilities of a model in simulating meteorological phenomena are an important research direction but beyond the scope of this study. Therefore, we would like to contribute some opinions about the capabilities of WRF ARW/Noah LSM/SLUCM in simulating meteorological phenomena to urban climate modellers for their reference.

At first, a reasonable temporal variation of the height of PBL can be seen in the modelled results but it cannot be examined by the observation because of unavailability of its corresponding observed ones. Secondly, the land-sea breeze should can be observed in our study area because it is a coastal area. However, we didn't find out this phenomenon in the modelled results due to the temporal resolution of 6 hours isn't enough fine for supporting the investigation of it (only four modelled variables of 10-meters wind speed at 2:00, 8:00, 14:00 and 20:00 on each day). Thirdly, the annual climatological variation in our study area is associated with the monsoon flow, especially the annual variations of 2-meters air temperature, 10-meters wind speed and precipitation. Figures 2, 13 and 16 demonstrate that the modelled 2-meters air temperature, 10-meters wind speed and precipitation have a same annual variation behaviour as its corresponding observed ones, which indicates that the model can simulate these climatological features in a study area affected by the monsoon flows. However, the model cannot reach the extreme value of these variables, especially the precipitation. Finally, the spatial distribution of temperature is strongly associated with the local land surface attributes. The model can simulate the temperature difference between in urban and non-urban area (Figure 8).

[revised manuscript text omitted]

**Diurnal Variation of Air Temperature**

**Figure 4: Diurnal Variation of 2-Meters Air Temperature.**

[Figure]

**Figure 5: Monthly PDF of 2-Meters Air Temperature Difference.**

[Figure]

**Figure 6: Monthly PSS of Surface Temperature at 2:00.**

[Figure]

**Figure 7: Monthly PSS of Surface Temperature at 14:00.**

[Figure]

**Figure 8: Comparison of Modelled Surface Temperatures with its Corresponding MODIS Ones at 2:00 and 14:00.**

[Figure]

**Figure 9: Comparison of MODIS Surface Temperatures in Urban Area with the Ones in the Non-Urban area at 2:00 and 14:00.**

[Figure]

**Figure 10: Comparison of Modelled Surface Temperatures in Urban Area with the Ones in the Non-Urban area at 2:00 and 14:00.**

[Figure]

**Figure 11: Monthly PDF of 2:00 Surface Temperature Difference.**

[Figure]

**Figure 12: Monthly PDF of 14:00 Surface Temperature Difference.**

[Figure]

**Figure 13: Monthly PSS of 10-Meters Wind Speed.**

[Figure]

[Figure]

[Figure]

**Figure 14: Comparison of Modelled 10-Meters Wind Speed with its Corresponding Observed Ones at 2:00, 8:00, 14:00 and 20:00.**

[Figure]

**Figure 15: Monthly PDF of 10-Meters Wind Speed Difference.**

[Figure]

**Figure 16: Monthly PSS of Precipitation.**

[Figure]

**Figure 17: Comparison of Modelled Precipitations with its Corresponding Observed Ones at 2:00, 8:00, 14:00, and 20:00.**

[Figure]

5    **Figure 18: Monthly PDF of Precipitation Difference.**

[Figure]

**Figure 19: Monthly PSS of Relative Humidity.**

[Figure]

5    **Figure 20: Comparison of Modelled Relative Humidity with its Corresponding Observed One at 2:00, 8:00, 14:00, and 20:00.**

[Figure]

**Figure 21: Monthly PDF of Humidity Difference.**

---

## Author Comment (AC7) · 22 Jan 2019

Response to Reviewer 2 [Cover Letter] Dear Reviewer, We appreciate your precious time in reviewing our paper and valuable comments. It is your valuable and insightful comments that led to possible improvements in the current version. The authors have carefully considered the comments and tried our best to address every one of them. We hope the revisions meet your high standards. The authors welcome further constructive comments if any. We provided the point-by-point response as below. Modifications in the manuscript are highlighted in red.

Sincerely,

Bo Huang, PhD

bohuang@cuhk.edu.hk Professor, Department of Geography and Resource Management The Chinese University of Hong Kong

[General Comment] The paper addresses the importance of model evaluation and presents a robust method for evaluating the results from urban climate simulations. Overall, the paper is clear and well structured. The discussion on natural gap, observation bias and model bias is substantial, highlighting the problems existing in current modeling practices that the climatological modelers should pay more attention to. The study is valuable to be published in a high impact journal. I would suggest a minor revision in which the authors should focus more on evaluation framework and clarify some technical points.

Response: The reviewer made constructive comments to improve the presentation and structure of the paper. We better organized the presentation of the proposed evaluation framework by including a clear workflow of the evaluation framework, more justification of PSS theory and other tools, and a summary table of evaluation results in our case study. With respect to the definition of 'acceptable', we discussed it and summarized a framework of the practical grading guidelines, which shall be further refined given the proposed evaluation tools being applied in many other case studies. Moreover, thanks for your interest in the developed high-resolution urban surface data. We are preparing another paper on it in which we compared the modeling results using the coarse urban land surface data provided by the WRF ARW model and the newly developed high-resolution urban land surface data. The paper should come out soon. Furthermore, regarding the selection of schemes for the physics components, provided more details in the later version.

Major Comments: [Comment 1] 1. The focus of this paper should be the model evaluation. The authors may strengthen the introduction and discussion of the evaluation framework in the following aspects: (1) Presentation of the evaluation framework: the

authors should summarize and present the evaluation framework in a visualized and more straightforward way (for example, using a workflow diagram).

Response: Thank you for your suggestions. We added a detailed explanaion of the proposed model evaluation framework in Section 2.2 [Pg3, Ln24 – Pg4, Ln5]. We added Table 1 for better presentation of the proposed model evaluation framework, in which we included three temporal persepectives, entire period, monthly, and daily, and two groups of tools, descriptive statistics and statistical distributions. Moreover, we presented the workflow for model evaluation in Section 2.4 [Pg5, Ln2 – Pg6, Ln7]. We hope the extended explanations made the proposed framwork easier to understand.

(2) Justification for the evaluation tools: the authors should introduce more PSS theory and explain why it is suitable to evaluate the model for urban climate simulations. The same as the PDF analysis and other evaluation tools.

Response: The importance of examining climate statistics other than climate means is not new (Katz and Brown 1992; Boer and Lambert 2001). The descriptive statistics are useful in providing aggregated information on the distribution of the attributes, but they can be very misleading since very different distributions can lead to similar descriptive statistics, and these aggregated metrics can be sensitive to outliers. Therefore, we examine not only the descriptive statistics but also metrics regarding the statistical distributions of modeled and observed meteorological attributes. The advantages of PDF and PSS for climate statistics have also been discussed by Perkins et al. (2007). We have also updated the manuscript accordingly in Section 2.2 [Pg3, Ln24 – Pg4, Ln5].

(3) Interpretation of the evaluation results: the authors kept using "acceptable" to describe the results. But how to define "acceptable"? What is the value of PSS would be considered as "not acceptable"? To make it a complete framework, the authors should provide guidelines to evaluate the results from the model evaluation.

Response: Thanks for your great suggestions. We agree that a reasonable definition

of "acceptable" would improve the novelty of this manuscript. We summerized the 72 monthly analysis for 6 meteorological attributes in our case study. The PSS values generally followed a normal distribution ranging from 0.444 to 0.886 with an average of 0.660 and a standard deviation of 0.098. Therefore, the PSS values are larger than 0.500 with a probability of 95%.

Figure 1. Histogram of the Perkins skill score for 72 monthly PDF analysis. A normal distribution was fit. The red dashed line indicate the lower bound for 95% confidence (PSS=0.500). Based on our results, we define 2 criteria for "acceptable" high-resolution urban climate simulations: 1) Yearly average PSS $\geq$ 0.550; 2) For each meteorological attribute, PSS $\geq$ 0.500 with a confidence interval of 95%. Compared to the case studies in Perkins et al. (2007), the lower bounds for PSS in our standard was lower, which is due to the increased resolution in our simulations. We are fully aware that, despite the sophisticated analysis we have conducted on different spatial and temporal scales, it is difficult to define 'acceptable' using results from one case study. In this vein, the proposed standard of 'acceptable' for high-resolution urban climate simulations based on our case study was meant to be the starting point and to be improved by future case studies using the proposed model evaluation framework. We added a guideline for PSS grading in Section 2.4 [Pg5, Ln2 – Pg6, Ln7].

(4) Intervals in PDF analysis: the authors use intervals of [-1, 1], [-2, 2], [-3, 3] for all variables in the PDF analysis. However, the significance of 3 degree in temperature change should have higher impact than 3 millimeter in precipitation. The authors should consider how to choose reasonable intervals for different variables.

Response: Thank you very much for your comment. Indeed, the same intervals have very different meanings for different meteorological attributes. Therefore, we used the standard deviations as intervals of the PDF analysis instead of fixed intervals. Moreover, we provided a guideline for specifying the interval in Section 2.4 [Pg5, Ln2 – Pg6, Ln7].

(5) Selection of variables: the authors should state the rationale for choosing variables for model evaluation in your case study.

Response: We included all meteorological variables that are meaningful for urban climate analysis in the model evalutaion. We added a explanation in Section 2.3 [Pg4, Ln8 – Pg4, Ln19].

(6) Next steps: the authors should discuss the drawbacks of the proposed evaluation framework and provide suggestions for future research. It would be a plus if the authors provide the source codes and original datasets using in the model evaluation.

Response: Thank you for your comment. We added more discussions on drawback of this study and possible future directions in the Section 5.4 [Pg12, Ln15-18]. We are planning to open the entire dataset in another paper specifically on the development of the high-resolution urban land surface data under review.

[Comment 2] 2. Although the inputs and setups in the modeling are critical to the model results, however, they are not the emphasis for this paper, and thus the modeling details should be listed in the appendix. On the other hand, a table of summarizing the evaluation results should be presented.

Response: Thank you for your advise. Regarding the modeling details (Section 3), we agree with comments from a previous reviewer that modeling details such as the input data processing and physical schemes used in our simulation are necessary to be included in the manuscript since they could have a significant impact on the simulation results. Therefore, we kept brief modeling details in Section 3 and included more in Section S4 of Supplementary Material.

[Comment 3] 3. Here are some suggestions the authors may take into consideration for their future research by applying their proposed evaluation method in investigating the model components and setups.

Response: Thank you very much for your suggestions. We definitely agree with you

that many more can be done and we will continue to thrive in this direction. We added some ideas for future research in Section 5.4 [Pg12, Ln15-18].

(1) New developed urban data: the authors developed four new sets of high-resolution urban data for modeling urban climate. What impact they have on the model results? Do they improve the overall performance of the model? If so, how much the improvement?

Response: Thank you for your comment. That is a great point for future research. Acturally, we have another manuscript under review focused on the development of the high-resolution urban land surface dataset and its effects on the reliability of climate simulation. Please refer to Archive. We can briefly introduce results from the other paper. People would naturally expect more accuracte modeling results with more accuracte input data. Unexpectedly , we find that high-resolution urban land surface datasets could either increase or decrease the evaluated reliability of simulation results, which is probably why not all modelers refine the land surface input data before the simulation. We believe the reason for this phenomenon is due to inperfect model and inperfect model evaluation methods. First, imperfectness in the detailed physical processes included in the model. This is the root why more accurate input data does not necessarily lead to more accurate simulation results. Second, it's inperfect to compare the grid-based simulation results with point-based observations. Moreover, the evaluated reliability cannot be compared across scales since high-resolution simulation contain many more details and will naturally decrease the evaluated reliability. Nevertheless, even the decrease in model evaluation metrics does not mean that the simulation results are less accuracte. We argue that providing more accurate input data into the model is the only way to prevent the 'garbage in, garbage out' effect, motivate us to refine the model itself and model evaluation methods, and lead us towards better modeling practice.

(2) Schemes of physics components: How to choose the schemes for each component? Would the selection of schemes have impacts on PSS scores?

Response: That is another great point for future research. The interactions among se-
lected physical components are very complex and so it can be difficult to provide solid
foundations for the selections. One way out is to compare the accuracy of simulation
results using different combinations of physical components. However, there are three
possible challenges: 1) many variables are involved in this process, including physical
components, model parameters, and spatial-temporal resolutions, which makes a very
large solution space. It would be compotationally expensive to try every combination;
2) Evidence from a certain study area and time period may not be transferable to other
study areas or time periods; 3) proper model evaluation metric. Comparing grid-based
modeling results with point-based simulations are naturally biased. Better evaluaed
accuracy does not necessarily mean better quality of simulation results. Therefore, we
think some evidence can definitely be provided from experiments using specific study
area, time period, model parameter, and spatial-temporal resolution. But it would be
difficult to provide more general insights for the seleciton of physical components.
Theoretical discussions and the 'try and error' process are still vital in refining such
selections.

Please also note the supplement to this comment:
https://www.geosci-model-dev-discuss.net/gmd-2018-220/gmd-2018-220-AC7-
supplement.pdf
* * *
[Figure]

Figure 1. Histogram of the Perkins skill score for 72 monthly PDF _analysis_. A normal distribution was fit. The red dashed line _indicate_ the lower bound for 95% confidence (PSS=0.500).

**Fig. 1.**

**Supplement:**

[revised manuscript text omitted]
 wasn't provided in the previous literatures. It is especially a research gap in urban climate modelling community to proposing a systematic framework and methods for model evaluation. Thus, in this paper, we dig deeply into the model evaluation and propose a systematic framework and methods for evaluating model results from multiple perspectives, to benefit future studies with more choices for model quality control and make urban scale simulation more robust. Moreover, we also provide a case analysis of the interval between the modelled atmospheric variable and its corresponding observed one.

35

The remainder of this paper is organized as follows. Section 2 introduces the proposed framework for model evaluation, experimental design, and data used for modelling and model evaluation. Section 3 introduces the technical preparation for the urban climate simulation. Section 4 presents various results of the proposed model evaluation methods in our case study. Section 5 concludes the paper with discussions.

**2 Methodology**

30

**2.1 Urban Climate Modelling**

[revised manuscript text omitted]

| Metrics                     |                                                                                            | Temporal Perspectives                             |                                                             |                 |
|-----------------------------|--------------------------------------------------------------------------------------------|---------------------------------------------------|-------------------------------------------------------------|-----------------|
| Statistical
Perspectives | Method                                                                                     | Annual                                            | Monthly                                                     | Daily           |
| Descriptive Statistics      | Temporal comparison                                                                        | Annual Pattern                                    | Monthly Pattern                                             | Diurnal pattern |
| Statistical                 | Overlap of the probability
density functions of
modelled and observed
data        | Annual mean PSS                                   | Monthly PSS                                                 |                 |
|                             | Probability density
function of the difference
between modelled and
observed data | Annual Mean
score of the
specified interval | Distribution and
the score of the
specified intervals |                 |

**2.3 Observation Datasets and Modelled Variables for Model Evaluation**

In existing literatures, Numerical Weather Prediction (NWP) models are typically evaluated by comparing the spatialtemporal patterns of the modelled variables with those of its corresponding near-surface observations. Moreover, four prognostic variables (2-meter air temperature, surface temperature, 10-meter wind at u direction and 10-meters wind at v

10

15

direction) and three diagnostic variables (accumulated total cumulus precipitation, accumulated total grid precipitation and 2-meter relative humidity) were chosen as the modelled variables for the comparison because these variables are the critical variables in the prognostic and diagnostic equations in the NWP model.

Table 2 shows the observation datasets we used for the comparison between modelled results and observations in the innermost domain. Table 3 lists the modelled variables and its corresponding observed ones in the model evaluation. The observation datasets are the point data except the MODIS dataset which is the grid data. All the modelled variables are grid data. The comparisons between modelled variables with its corresponding observed ones are comparisons between the grid value of the modelled variable with its point value matched by to geographical locations, except for the comparison between

**Table 2: Observation Datasets.**

| Observation Datasets                | Sources                                         |  |
|--------------------------------------------|-------------------------------------------------|--|
| 2010 PRD 2-Meters Air Temperature          |                                                 |  |
| 2010 PRD 10-Meters Wind Speed              | Meteorological Bureau of Shenzhen Municipality  |  |
| 2010 PRD Precipitation                     | increation pareau of Shenzhen Municipanty       |  |
| 2010 PRD Relative Humidity                 |                                                 |  |
| 2010 MODIS/Aqua Land Surface               | NASA EOSDIS Land Processes DAAC, USGS Earth     |  |
| Temperature and Emissivity (LST/E) product | Resources Observation and Science (EROS) Center |  |

the modelled surface temperature and its corresponding observation retrieved from MODIS imagery.

| Modelled Variables for Model Evaluation |                                            | Corresponding Observation Datasets       |  |
|-----------------------------------------|--------------------------------------------|------------------------------------------|--|
| Name                                    | Description                                | Corresponding Observation Datasets       |  |
| T2                                      | 2-meters air temperature                   | 2010 PRD 2-Meters Air Temperature        |  |
| TSK                                     | Surface temperature                        | 2010 MODIS/Aqua Land Surface Temperature |  |
|                                         |                                            | and Emissivity (LST/E) product           |  |
| U10                                     | 10-meters wind at U direction              | 2010 PRD 10-Meters Wind Speed            |  |
| V10                                     | 10-meters wind at V direction              |                                          |  |
| RAINC                                   | Accumulated total cumulus precipitation    | 2010 PRD Precipitation                   |  |
| RAINNC                                  | Accumulated total grid scale precipitation |                                          |  |
| RH2                                     | 2-meters relative humidity                 | 2010 PRD Relative Humidity               |  |

**2.4 Workflow for Model Evaluation**

Following the proposed framework, we used four statistic figures (Table 4) and designed a workflow (Figure 1) in the practice of model evaluation. We first conducted a data processing operation (run by a program) on each pair of the raw
observation dataset and its corresponding modelled result to produce an evaluation 3D-matrix which consists of a one-year temporal series of 2D-matrixes. Each non-empty element of the 2D-matrixes geographically corresponds with a pair of observation data and its corresponding modelled one a time point. A location map of meteorological observations was also produced if the raw observation dataset is a meteorological observation dataset. Secondly, we conducted the figure plotting operations (run by the programs) on each evaluation 3D-matrix step by step to produce a series of statistic figures. Specially,

10 we designed a guideline (Table 5) for specifying the intervals in the PDFD figure, which are used for measuring the accuracy. Finally, we conducted grading or checking operations on these statistic figures artificially base on the guidelines for grading (Table 6) or checking (Table 7).

| Statistical                  | Methods                                            |                                                                                                                                                                                                                                                                                 |
|------------------------------|----------------------------------------------------|---------------------------------------------------------------------------------------------------------------------------------------------------------------------------------------------------------------------------------------------------------------------------------|
| Perspectives                 | Statistic Figures                                  | Usage                                                                                                                                                                                                                                                                           |
| Descriptive                  | Temporal Comparison of
Spatial Variation (TCSV) | It is used to temporally compares two variables' spatial variation
ranges and median in the whole year.                                                                                                                                                                      |
| Statistics                   | Diurnal Variation (DV)                             | It is used to temporally compares the diurnal variations of two
variables' spatial variation ranges and median in the whole year.                                                                                                                                            |
|                              | ProbabilityDensityFunctionofDifference(PDFD)       | It is used to show the probability density of difference between
the modeled variable and its corresponding observed one.                                                                                                                                                    |
| Statistical
Distributions | Perkins Skill Score (PSS)                          | It is used for revealing quantifiably the extent of overlap
between the observed and modeled variables' Probability
Density Function (PDF). A value of 1 indicates a perfect
modeling of the observation. On the contrary, a value of 0 means
the worst simulation. |

**Table 4: The Statistic Figures for Model Evaluation.**

15 Table 5: The Guideline for Specifying the Interval in a PDFD Figure.

| No. | The Range of Coefficient (A) | Intervals             |
|-----|------------------------------|-----------------------|
| 1   | 0.1 – 0.2                    |                       |
| 2   | 0.2 – 0.4                    | $[-A\sigma, A\sigma]$ |
| 3   | 0.4 – 0.6                    |                       |

Remark:  $\sigma$  is the annual mean value of the monthly standard deviation of the modelled variable

**Table 6: The Guideline for Grading.**

| Ra                  | Grading                |              |
|---------------------|------------------------|--------------|
| PSS                 | Accuracy(a)            | Oraung       |
| $0.7 \le PSS \le 1$ | $70\% \le a \le 100\%$ | Good         |
| $0.7 > PSS \ge 0.5$ | $70\% > a \ge 50\%$    | Acceptable   |
| PSS < 0.5           | a < 50%                | Unacceptable |

Remark: Accuracy is the PDFD value of interval 2 or interval 3.

5

**Table 7: The Guideline for Checking.**

| Statistic | Temporal Perspectives for Checking |                 |                 |
|-----------|------------------------------------|-----------------|-----------------|
| Figures   | Annual                             | Monthly         | Daily           |
| TCSV      | Annual Pattern                     | Monthly Pattern |                 |
| DV        | Annual Pattern                     |                 | Diurnal pattern |

**3** Technical Preparation**

**3.1 Model Setup**

10 A telescoping nests' structure with four nested domains which are centered at 22°39'30" N, 114°11'30", was set up as the horizontal domain baseline configuration in this study. Moreover, the same set of eta levels with 51 members was used in each horizontal domain. Furthermore, there were some physics components in the model, and each component had some different schemes for choosing. Table 7 shows the scheme chosen for each component. For more details, please refer to Section S4 of Supplementary Material.

**15 Table 7: Physics Components' Schemes.**

| Component                | Scheme                          |
|--------------------------|---------------------------------|
| Cumulus                  | New Simplified Arakawa-Schubert |
| Microphysics             | WDM5                            |
| Radiation                | RRTMG                           |
| Planetary Boundary Layer | Bougeault-Lacarrere             |
| Surface Layer            | Revised MM5                     |
| Land Surface Model       | Noah LSM                        |
| Urban Canopy Model       | Single-layer                    |

**3.2 Data Preparation**

Firstly, the 2010 NCEP FNL (Final) Operational Global Analysis Dataset (1-degree grid spatial resolution and 6-hourly temporal resolution) was used as the Gridded Data in this study. Secondly, the Completed Dataset of WRF Preprocessing

System (WPS) Geographical Input Data was used as the Static Geographical Dataset in this study. Thirdly, the 2010 PRD Urban Land Surface Dataset, whose major sets of data include the land cover, vegetation coverage, urban morphology and anthropogenic heat, was specially developed for refining the WRF primary data.

**3.3 Primary Data Processing**

5 Firstly, the primary data included the interpolated geo-data files, the intermediate format meteorological data files, the horizontally interpolated meteorological data files, the initial condition data files, and the lateral boundary condition data files. Secondly, two primary data processing software packages (geo\_data\_refinement processing package and wrf\_input\_refinement processing package) were developed for extracting the urban land surface attributes from the 2010 PRD Urban Land Surface Dataset and revising the corresponding fields of the related primary data files with these attributes.

**10 4 Model Evaluation**

15

**4.1 Evaluation of the 2-Meters Air Temperature**

As shown in Figure 2, the monthly PSS of 2-meters air temperature ranges from a minimum of 0.595 in July to a maximum of 0.886 in January and has an annual mean value of 0.724. This makes it clear that the model captured the PDF for the observed air temperature at least about 60% in a month and over 72% in a year. Moreover, Figure 3 shows monthly comparisons between the observed and the modelled 2-meters air temperatures' spatial variation range and median values at 2:00, 8:00, 14:00, and 20:00. It is evident that the modelled air temperatures always have similar behaviour in temporal-

- spatial variation with the observed ones. Furthermore, Figure 4 shows the diurnal variations of observed, modelled air temperatures' median and spatial variation range in each month. As is evident in Figure 4, both the median and the range of the 2-meters modelled air temperature have the same diurnal variation pattern as that of its corresponding observed ones in
  20 each month, although there are differences between the modelled ones and the corresponding observed ones. Finally, Figure 5, block and the DDE of block are discussed on the same diagonal data and the corresponding observed ones. Finally, Figure 5, block are discussed on the same diagonal data and the corresponding observed ones. Finally, Figure 5, block are discussed on the same diagonal data are discussed on the same data are dat

[revised manuscript text omitted]

**5.3 The Capabilities of WRF ARW/Noah LSM/SLUCM in Simulating Meteorological Phenomena**

This paper aims to present a standardized methodological framework that can be used in all regions to evaluate the 10 modelling performance rather than to examine the capabilities of the model in simulating meteorological phenomena in a specified region. Therefore, we proposed model evaluation methods for the comparison between modelled variables and its corresponding observed ones rather than investigating the meteorological phenomena in the modelled results. However, the capabilities of a model in simulating meteorological phenomena are an important research direction but beyond the scope of this study. Therefore, we would like to contribute some opinions about the capabilities of WRF ARW/Noah LSM/SLUCM 15 in simulating meteorological phenomena to urban climate modellers for their reference.

- At first, a reasonable temporal variation of the height of PBL can be seen in the modelled results but it cannot be examined by the observation because of unavailability of its corresponding observed ones. Secondly, the land-sea breeze should can be observed in our study area because it is a coastal area. However, we didn't find out this phenomenon in the modelled results due to the temporal resolution of 6 hours isn't enough fine for supporting the investigation of it (only four modelled variables
- 20 of 10-meters wind speed at 2:00, 8:00, 14:00 and 20:00 on each day). Thirdly, the annual climatological variation in our study area is associated with the monsoon flow, especially the annual variations of 2-meters air temperature, 10-meters wind speed and precipitation. Figures 2, 13 and 16 demonstrate that the modelled 2-meters air temperature, 10-meters wind speed and precipitation have a same annual variation behaviour as its corresponding observed ones, which indicates that the model can simulate these climatological features in a study area affected by the monsoon flows. However, the model cannot reach
- the extreme value of these variables, especially the precipitation. Finally, the spatial distribution of temperature is strongly 25 associated with the local land surface attributes. The model can simulate the temperature difference between in urban and non-urban area (Figure 8).

[revised manuscript text omitted]

---

## Author Comment (AC9) · 14 May 2019

**Supplementary Material**

**S1 The Urban Climate Modelling System**

We used the Advanced Research WRF (ARW) modelling system coupled with Noah LSM/UCM Model (Noah Land Surface Model/Urban Canopy Model) developed by National Center of Atmospheric Research in this study for the urban climate simulation. It is a limited-area, non-hydrostatic and meso-scale atmospheric modelling system with the terrain-following mass vertical coordinate, designed for atmospheric research applications (Skamarock et al., 2005, 2008; Lo et al., 2008). The ARW model is a typical atmospheric model integrating with a set of five interacting physical components (Microphysics, Cumulus Parameterization, Radiation, Planetary Boundary Layer/Vertical Diffusion and Surface) (Skamarock et al., 2005, 2008).

The Noah LSM model is coupled with the ARW model by the surface component. The in-homogeneity of the surface affects energy and mass redistribution in the atmosphere. The Noah LSM model utilizes the following parameters in the representation of the in-homogeneous texture of the surface to simulate the land surface process.
- Land use
- Land covers (vegetation)
- Soil texture
- Secondary parameters related to the above three primary parameters

**S2 The 2010 PRD Urban Land Surface Dataset**

The 2010 PRD Urban Land Surface Dataset includes the land cover data, vegetation coverage data, urban morphology data, and anthropogenic heat data, for which the spatial resolution is 1-km$^2$. The vegetation coverage includes 12 monthly vegetation coverage maps. The urban morphology data includes the urban fraction, the fraction of building area, the mean building height area weighted, the building surface area to plan area ratio, the mean building height, the standard deviation of mean building height area weighted, and the frontal area index. The anthropogenic heat data includes the anthropogenic sensible heat and the anthropogenic latent heat.

**S3 Design of the Four-days Simulation Segment**

An atmosphere model initially needs to run for a period of time in order to stabilizing its own condition. Modelling results during this period is frustrating. Normally, to reduce the negative effect of model instability in the initial model run, the model result data is discarded. This procedure is called model spin-up. However, there is no statistical report told the modellers how long the spin-up

5 for running a model is the best (Kleczek et al. 2014). Nevertheless, a minimum spin-up time of 12 hours is necessary for balancing the NWP model (Jankov et al. 2007; Skamarock and Klemp, 2008; Kleczek et al. 2014). Moreover, practically in the atmosphere modeller community that the longer a simulation period is, the longer of the spin-up time is required. Similarly, if the period of a weather simulation case is too long, the results of the last few days might be distorted. Therefore, the appropriate simulation and spin-up periods would improve the quality of the result in a weather simulation case. To sum up, the period of a simulation segment

10 was set to 4 days and the first day was used for the spin-up period.

In this study, one-year urban climate simulation case was divided into a series of sequent simulation segments. The first day of the next simulation segment overlaps with the last day of the previous simulation segment. The sequence of simulation segments for an urban climate simulation case is shown in Figure S1.

[Figure]

**Figure S1: The Simulation Segments' Sequence.**

**S4 Schemes Choosing for Physics Components of WRF ARW/Noah LSM/SLUCM**

There are 7 physical components in the model, and each component has different candidate schemes.

(1) Cumulus Parameterization

(2) Microphysics

5 (3) Radiation

(4) Planetary Boundary Layer

(5) Surface layer

(6) Land Surface Model

(7) Urban Canopy Model

First, Cumulus Parameterization directly outputs the effects of physics process rather than simulates the physics process itself (Chen, 2011). The New Simplified Arakawa-Schubert scheme was chosen for Cumulus Parameterization, which supports deep and shallow convection and momentum transport for the ARW core.

15 Second, the microphysics component is responsible for the processes of resolved water, cloud and precipitation (Skamarock et al., 2005, 2008). Based on the sophistication ranking of the schemes, the WDM5 scheme was chosen for all domains in this study.

Third, the Radiation component simulates the atmospheric radiation processes in a vertical column of a horizontal grid. It consists of longwave and shortwave modules and uses the precipitation, water vapour and cloud-related variables as the inputs. It also

20 exchanges the radiation fluxes related variables with the surface component and updates the potential temperature-related variables. The RRTMG scheme also was chosen as the shortwave and longwave radiation scheme because it supports the climatological ozone and aerosol data input which interacts with microphysics and cumulus parameterization components by $Q_c$, $Q_r$, $Q_i$, and $Q_s$.

Fourth, the land-surface models (LSMs) requires the input of the atmospheric data from surface layer component, the downward

25 radiative fluxes from radiation component and the precipitation data from microphysics and cumulus parameterization components. It outputs the heat and moisture fluxes data to PBL component and the upward radiative fluxes to radiation component over land and sea-ice points (Skamarock et al., 2008). The Noah LSM was chosen as the scheme of land-surface model for the inner-most horizontal domain because the urban canopy model can only be coupled with Noah LSM.

30 Fifth, the Urban Canopy Model component is responsible for the physical processes of land surface in urban environment. The Single Layer Urban Canopy Model was chosen as the scheme of the Urban Canopy Model because of its sophistication.

Sixth, the Planetary Boundary Layer component simulates the vertical heat, moisture and momentum fluxes vertical diffusion which are caused by the turbulence exchanges in a whole column of a grid (Wang, 2014, 2015). The Bougeault–Lacarrere scheme was chosen for all domains because it supports the NUDAPT format data that used in this study for taking the urban morphology into account.

Seventh, in the WRF model, the Surface Layer component is responsible for providing the friction stress, the surface fluxes of heat and moisture to PBL (Skamarock et al., 2008). As a result of a comparison, the revised MM5 surface layer scheme was chosen for all horizontal domains in this study.

**S5 Guideline for Model Evaluation**

10   We used four statistic figures (Table S1) in the practice of model evaluation. We first conducted a data processing operation (run by a program) on each pair of the raw observation dataset and its corresponding modelled result to produce an evaluation 3D-matrix which consisted of a one-year temporal series of 2D-matrixes. Each non-empty element of the 2D-matrixes geographically corresponded with a pair of observation data and its corresponding modelled one at a time point. A location map of meteorological observations was also produced if the raw observation dataset was a meteorological observation dataset. Secondly, we conducted

15   the figure plotting operations (using the programs) on each evaluation 3D-matrix step by step to produce a series of statistic figures. Specially, we designed a guideline (Table S2) for specifying the intervals in the PDFD figure, which were used for measuring the accuracy. Finally, we conducted grading or checking operations on these statistic figures artificially based on the guidelines for grading (Table S3) or checking (Table S4).

Table S1: The Statistic Figures for Model Evaluation.

| Statistical Perspectives | Methods | |
|---|---|---|
| | Statistic Figures | Usage |
| Descriptive Statistics | Temporal Comparison of Spatial Variation (TCSV) | It was used to temporally compare two variables' spatial variation ranges and median in the whole year. |
| | Diurnal Variation (DV) | It was used to temporally compare the diurnal variations of two variables' spatial variation ranges and median in the whole year. |
| Statistical Distributions | Probability Density Function of Difference (PDFD) | It was used to show the probability density of difference between the modelled variable and its corresponding observed one. |
| | Perkins Skill Score (PSS) | It was used for revealing quantifiably the extent of overlap between the observed and modelled variables' Probability Density Function (PDF). |

| | | A value of 1 indicated a perfect modelling of the observation. On the contrary, a value of 0 meant the worst simulation. |
|---|---|---|

**Table S2: The Guideline for Specifying the Interval in a PDFD Figure.**

| No. | The Range of Coefficient (A) | Intervals |
|-----|------------------------------|-----------|
| 1 | 0.1 – 0.2 | |
| 2 | 0.2 – 0.4 | [-Aσ, Aσ] |
| 3 | 0.4 – 0.6 | |

**Remark: σ is the annual mean value of the monthly standard deviation of the modelled variable**

**Table S3: The Guideline for Grading.**

| Range | | Grading |
|-------|--|---------|
| PSS | Accuracy(a) | |
| $0.7 \leq PSS \leq 1$ | $70\% \leq a \leq 100\%$ | Good |
| $0.7 > PSS \geq 0.5$ | $70\% > a \geq 50\%$ | Acceptable |
| $PSS < 0.5$ | $a < 50\%$ | Unacceptable |

**Remark: Accuracy is the PDFD value of interval 2 or interval 3.**

**Table S4: The Guideline for Checking.**

| Statistic Figures | Temporal Perspectives for Checking | | |
|-------------------|--------|---------|-------|
| | Annual | Monthly | Daily |
| TCSV | Annual Pattern | Monthly Pattern | |
| DV | Annual Pattern | | Diurnal pattern |

**S6 Observation Datasets**

A quality control had been applied to all meteorological observation datasets by the data provider. Table S1 shows the total number of observations and the numbers of observations in urban area and non-urban area. Moreover, Figures S2, S3, S4 and S5 show the locations of the meteorological observations.

**Table S1: The Numbers of Meteorological Observations**

| Type | Number in urban area | Number in non-urban area | Total number |
|------|----------------------|--------------------------|--------------|
| 2-meters temperature | 34 | 23 | 57 |

| 10-meters wind speed | 37 | 26 | 62 |
|---|---|---|---|
| Precipitation | 31 | 33 | 64 |
| Relative humidity | 15 | 9 | 24 |

[Figure]

**Figure S2: Temperature Observations in Domain 4.**

[Figure]

**Figure S3: The 10-Meters Wind Observations in Domain 4.**

[Figure]

**Figure S4: Precipitation Observations in domain 4.**

[Figure]

**Figure S5: Relative Humidity Observations in domain 4.**

5    The MODIS/Aqua Land Surface Temperature and Emissivity (LST/E) product (Short name: MYD11A1) provided by the U.S. Geological Survey (USGS) was used for the evaluation. This product includes a grid surface temperature with 1-km horizontal resolution at around 2:00 and 14:00 (Beijing time) per day. It also has a quality control attribute for each surface temperature record

to identify the level of data quality. Such quality control attribute was used for filtering the poor-quality records that was at least 5 degrees' different from the corresponding modelled value.

**S7 Figures for Comparisons in Surface Temperature, 10-Meters Wind Speed, Precipitation and Relative Humidity**

[Figure]

5    **Figure S6: Comparison of Modelled and MODIS Surface Temperatures at 2:00 and 14:00.**

[Figure]

**Figure S7: Comparison of MODIS Surface Temperatures (at 2:00 and 14:00) in Urban and Non-Urban Areas.**

[Figure]

**Figure S8: Comparison of Modelled and Observed Surface Temperatures (at 2:00 and 14:00) in Urban Area and the Non-Urban area.**

[Figure]

**Figure S9: Monthly PSS of Surface Temperature at 2:00.**

[Figure]

**Figure S10: Monthly PSS of Surface Temperature at 14:00.**

[Figure]

**Figure S11: Monthly PDF of 2:00 Surface Temperature Difference.**

[Figure]

**Figure S12: Monthly PDF of 14:00 Surface Temperature Difference.**

[Figure]

[Figure]

[Figure]

[Figure]

**Figure S13: Comparisons of Modelled and Observed 10-Meters Wind Speed at 2:00, 8:00, 14:00 and 20:00.**

[Figure]

[Figure]

[Figure]

[Figure]

**Figure S14: Comparison of Observed 10-Meters Wind (at 2:00, 8:00, 14:00 and 20:00) in Urban Area and in the Non-Urban area.**

[Figure]

[Figure]

[Figure]

**Figure S15: Comparison of WRF 10-meters Wind (at 2:00, 8:00, 14:00 and 20:00) in Urban Area and in the Non-Urban area.**

[Figure]

5    **Figure S16: Monthly PSS of 10-Meters Wind Speed.**

[Figure]

**Figure S17: Monthly PDF of 10-Meters Wind Speed Difference.**

[Figure]

[Figure]

[Figure]

[Figure]

**Figure S18: Comparison of Modelled and Observed Precipitations at 2:00, 8:00, 14:00, and 20:00.**

[Figure]

[Figure]

[Figure]

[Figure]

**Figure S19: Comparison of Observed Precipitation (at 2:00, 8:00, 14:00 and 20:00) in Urban Area and in the Non-Urban area.**

[Figure]

[Figure]

[Figure]

**Figure S20: Comparison of WRF Precipitation (at 2:00, 8:00, 14:00 and 20:00) in Urban Area and in the Non-Urban area.**

[Figure]

**Figure S21: Monthly PSS of Precipitation.**

[Figure]

**Figure S22: Monthly PDF of Precipitation Difference.**

[Figure]

[Figure]

[Figure]

[Figure]

**Figure S23: Comparison of Modelled and Observed Relative Humidity at 2:00, 8:00, 14:00, and 20:00.**

[Figure]

[Figure]

[Figure]

[Figure]

**Figure S24: Comparison of Observed Humidity (at 2:00, 8:00, 14:00 and 20:00) in Urban and in Non-Urban Areas.**

[Figure]

[Figure]

[Figure]

**Figure S25: Comparison of Modelled Relative Humidity (at 2:00, 8:00, 14:00 and 20:00) in Urban and Non-Urban Areas.**

[Figure]

**Figure S26: Monthly PSS of Relative Humidity.**

[Figure]

**Figure S27: Monthly PDF of Humidity Difference.**

**S8 The PSSs in Urban and Non-Urban Areas**

(a)                                                                          (b)

[Figure]

**Figure S28: Monthly PSS of 2-Meters Air Temperature in Urban (a) and Non-Urban (b) Area.**

(a)                                                                          (b)

[Figure]

Figure S29: Monthly PSS of Surface Temperature in Urban (a) and Non-Urban (b) Area.

(a)                                                                                         (b)

[Figure]

5    Figure S30: Monthly PSS of 10-Meters Wind Speed in Urban (a) and Non-Urban (b) Area.

(a)                                                                                         (b)

[Figure]

**Figure S31: Monthly PSS of Precipitation in Urban (a) and Non-Urban (b) Area.**

**(a)**                                                                                    **(b)**

[Figure]

5      **Figure S32: Monthly PSS of Relative Humidity in Urban (a) and Non-Urban (b) Area.**

**S9 The Capabilities of WRF ARW/Noah LSM/SLUCM in Simulating Meteorological   Phenomena**

This paper aims to present a standardized methodological framework that can be used in all regions to evaluate the modelling performance rather than to examine the capabilities of the model in simulating meteorological phenomena in a specified region. Therefore, we proposed model evaluation methods for the comparison between modelled variables and its corresponding observed

10    ones rather than investigating the meteorological phenomena in the modelled results. Nevertheless, the capabilities of a model in simulating meteorological phenomena are an important research direction but beyond the scope of this study.  Therefore, we would

like to contribute some opinions about the capabilities of WRF ARW/Noah LSM/SLUCM in simulating meteorological phenomena to urban climate modellers for their reference.

Firstly, a reasonable temporal variation of the height of PBL can be seen in the modelled results but it cannot be examined by the observation because of unavailability of its corresponding observed ones. Secondly, the land-sea breeze should be observed in our study area because it is a coastal area. However, we didn't find out this phenomenon in the modelled results due to the temporal resolution of 6 hours wasn't enough fine for supporting the investigation of it (only four modelled variables of 10-meters wind speed at 2:00, 8:00, 14:00 and 20:00 on each day). Thirdly, the annual climatological variation in our study area was associated with the monsoon flow, especially the annual variations of 2-meters air temperature, 10-meters wind speed and precipitation. Figures 2, 13 and 16 demonstrated that the modelled 2-meters air temperature, 10-meters wind speed and precipitation had the same annual variation behaviour as its corresponding observed ones, which indicated that the model can simulate these climatological features in a study area affected by the monsoon flows. However, the model cannot reach the extreme value of these variables, especially the precipitation. Finally, the spatial distribution of temperature was strongly associated with the local land surface attributes. The model can simulate the temperature difference between in urban and non-urban area.

---

## Author Comment (AC10) · 14 May 2019

**Response to Reviewers**

**[Cover Letter]**

Dear Editor,

We appreciate you and the reviewers for your precious time in reviewing our paper and providing valuable comments. It was your valuable and insightful comments that led to possible improvements in the current version. The authors have carefully considered the comments and tried our best to address every one of them. We hope the manuscript after careful revisions meet your high standards. The authors welcome further constructive comments if any.

Below we provide the point-by-point responses. All modifications in the manuscript have been highlighted in red.

Sincerely,

Bo Huang, PhD

bohuang@cuhk.edu.hk

Professor, Department of Geography and Resource Management

The Chinese University of Hong Kong

**Response to Reviewer 1**

**[General Comment]** The manuscript has been well improved, and I think most of my comments have been addressed with either new analysis or necessary discussions.

**Response:** Thank you very much.

Some minor revisions for the authors to consider:

**[Minor Comment 1]** P1L20-21 "...ambiguous or arbitrary to some extent, which may still lead to some seemingly new findings, but these findings may be scientifically misleading." Please consider rewriting this sentence.

**Response:** Thanks for your kind reminders. We revised the sentence as follows:

"Studies without systematic model evaluations, being ambiguous or arbitrary to some extent, may still lead to some seemingly-new but scientifically-misleading findings."[Pg1, Ln18-19]

**[Minor Comment 2]** P1L23 "urbanising city of Shenzhen, China...". Does the study area also include Hong Kong?

**Response:** Thanks for your kind reminders. We revised the sentence as follows[Pg1, Ln22]:

"To tackle these challenges, this article proposes a methodological framework for the model evaluation of high-resolution urban climate simulations and demonstrates its effectiveness with a case study in the area of Shenzhen and Hong Kong, China."

**[Minor Comment 3]** P2L30 Change "wasn't" to "has not been". Please also check through the text. For example, P11L19 "isn't".

**Response:** Thank you very much for the reminder. We have made revisions accordingly.

**[Minor Comment 4]** Please trimming Sect. 5.3.

**Response:** We moved Subsection 5.3 to Supplementary Material [Section S9].

**Response to Reviewer 2**

**[General Comment]** The revision highlights the importance of model evaluation and provides a reasonable approach for evaluating the results from urban climate simulations. Overall, the revision is well organized and clearly presented. In particular, Figure 1 is very nice and helpful to understand the framework. In my opinion this paper is worth to be published in GMD.

**Response:** Thank you very much for your previous comments that helped us improve this manuscript.

Minor suggestions:

**[Comment 1]** In Table 1, to use PDF & PSS, or probability density functions & Perkins Skill Score, instead of probability density functions & PSS;

**Response:** Revised accordingly.

**[Comment 2]** It would be better to put Table 4 & Figure 2-21 in the supplementary material.

**Response:** Revised accordingly.

**Response to Reviewer 3**

**[General Comment]** The overall intention of this submission is a good one. In general I agree that more attention should be paid to model evaluation in urban climate modelling work. However, this submission is relatively formulaic and it misses a number of key nuances that render the overall model assessment somewhat unconvincing and moreover unlikely to be applied by other researchers. I appreciate the authors' intent and believe more rigorous model evaluation is critically needed in the field. However, this submission does not currently approach GMD standards in my view.

**Response:** Thank you very much for agreeing with us to the intention of this manuscript. We have read your comments carefully and tried our best to address them one by one, especially in terms of providing a more rigorous model evaluation. We hope that the manuscript has been improved towards GMD standards after this revision.

**[Comment 1]** English grammar requires improvement throughout.

**Response:** We went through the entire manuscript to eliminate grammatical mistakes.

**[Comment 2]** There are too many tables. In general the presentation/communication requires improvement.

**Response:** Thank you for the nice reminder. We combined Tables 2 and 3 into one table (Table 2). Moreover, we moved Subsection 2.4 to Supplementary Material [Section S5] to cut down the number of tables.

**[Comment 3]** Figure captions are in general insufficiently detailed. For example, it is not clearly indicated whether central tendencies and variation in several figures are spatial or temporal.

**Response:** Thank you for your nice reminder. We revised most of the figure captions to make them clearer.

**[Comment 4]** P3L27-33: There is no mention of how spatial patterns are evaluated here. In general, it is not fully clear how spatial patterns produced by urban climate modelling at the city-regional scale is evaluated, which is a prime intention of these models.

**Response:** Thanks for your question.

At the spatial dimension, climatological studies usually focus on three scales, including local (less than $10^4$ km$^2$), regional (from $10^4$ to $10^7$ km$^2$) and global (greater than $10^7$ km$^2$) scales (Intergovernmental Panel on Climate Change, 2012). The similarity between the modelled and observed variables' spatial patterns is indeed a major content in model evaluation of

regional and global climate, especially, the spatial difference of precipitation belt and atmospheric circulation. In the previous literature, there were not many papers on the methods of spatial pattern comparison for local climate simulation, which is a research gap that we focused on.

In urban climatology, the most important spatial pattern of meteorological variables is the difference between in urban and non-urban areas. Therefore, at the spatial dimension, we evaluated the model using the temporal comparison of spatial variation in the whole year in urban and non-urban areas.

We added these contents into Subsections 2.2 [Pg4, Ln11-13], 4.1 [Pg6, Ln28 – Pg 7, Ln2; Pg10, Ln15] , 4.3 [Pg11, Ln27-29; Pg12, Ln3-4], 4.5 [Pg12, Ln24-27; Pg12, Ln34]and 5.4 [Pg15, Ln15-20]. Moreover, we added figures about the comparison of meteorological variables between urban and non-urban areas into the Supplementary Material (Figures S7, S8, S14, S15, S19, S20, S24, and S25).

**[Comment 5]** P4L9: 2m air temperature is diagnosed in WRF-SLUCM, and I believe that surface temperature is as well. 10-m wind is also diagnosed.

**Response:** Thank you very much for pointing this out. We revised the sentence as follows:

Pg4, Ln16-17: " Moreover, we chose 7 meteorological variables for the comparison, including 2-meter air temperature, surface temperature, 10-meter wind at u direction, 10-meters wind at v direction, accumulated total cumulus precipitation, accumulated total grid precipitation and 2-meter relative humidity, because these variables are the critical variables in the prognostic and diagnostic equations in the NWP model."[Pg5, Ln3-7]

**[Comment 6]** Table 3: TSK should not be compared to a MODIS land surface temperature product. TSK includes the emission of all building walls, whereas a satellite does not view longwave emission from any walls (nadir view), or only from walls of select orientations (off-nadir view). Many authors make this mistake; if this is to be a rigorous model evaluation paper that sets an example, it should not be made here. Instead, you must extract the surface temperatures of roads, roofs, and rural area and weight them yourself for each grid square. This is not necessarily trivial and requires adding an "h" to the appropriate variables in the WRF Registry as well as outputting the rural surface temperature (which may require adding a new variable and re-running the simulations).

**Response:** The reviewer missed a piece of important information - the scan angle of the MODIS satellite is ± 55°. At the end of a scan line, the scanning angle being 55° means that the land surface temperature retrieved from MODIS imagery is not an area-weighted mean value temperatures of roads, roofs, and rural area within a particular pixel. Actually, a MODIS land surface temperature is a result of the inverse calculation based on the longwave radiation through the atmosphere received by satellite according to the theory of blackbody. A MODIS land surface temperature is a  manifestation of the surface synthetic radiation brightness temperature.  In addition, in the land surface process, TSK is calculated iteratively

according to the energy balance which involves longwave radiation, shortwave radiation, sensible heat, and latent heat, and accordingly, the final TSK value is also a manifestation of the surface synthetic radiation brightness temperature. Although there are some differences between TSK and the brightness temperatures observed by satellites, they describe relatively similar physical quantities. Therefore, we used TSK to compare with MODIS land surface temperature. We also added more explanations into Subsection 2.3 [Pg5, Ln12-19].

[Comment 7] Figure 3: These results will depend entirely on how many measurement stations are available and included, will they not?

Response: We don't think so. Figure 3 is a figure of descriptive statistic. The results may only be distorted if the number of measurement stations is too small. Our statistical results were based on data from 57 stations, so we don't think these results will be very sensitive to the number of measurement stations included.

[Comment 8] P12L2-5: I disagree. I think they are both useful. In particular, with the use of the Perkins Skill Score, a climatological approach appears to be taken here. What is also important for urban climate modelling is that the temporal evolution of each variable is predicted accurately, particularly at diurnal time scales. For example, a measure of covariance between measurements and observation that includes many temporal data points within each cycle of forcing is important (e.g., hourly measurement-model comparison of air temperature would capture the covariance between measured and modelled diurnal variation).

Response: Thanks for your comment. We agree with you that temporal evolution is an important dimension of meteorological processes and should be included in the modeling evaluation. We did conduct comparisons of the temporal evolution, including comparisons of diurnal patterns in Figures 3, and comparisons of monthly patterns in Figure 2 and Figures S6, S13, S18 and S23 of Supplementary Material.

[Comment 9] Ultimately, I think model evaluation will always have a subjective element. How is the authors' method designed to be applicable to all regions, conditions, simulation durations and resolutions, etc?

Response: Thanks for your comment. Our study just focused on evaluating urban climate modelling. The purposes of model evaluations to different scale, conditions, simulation durations and resolutions is the same, which is to establish the trustworthiness of the modelling results. In the proposed framework, we demonstrate tools to compare both the descriptive statistics and the statistical distributions of the observed and simulated meteorological variables. We also demonstrate how to use the tools to compare the same meteorological attributes at different temporal aggregation levels, including daily and monthly/seasonal. For sure that case-specific adaptations have to be made before the proposed methods be applied to other regions, conditions, and resolutions, which is the same

for most models and methods. The simulation model itself has to be calibrated, with an enormous amount of efforts, to different areas as well. Of course, the proposed model evaluation methods are not perfect, but we are confident that our practice can serve well as a reminder and a guide that improves the urgently-lacking model evaluation in current practices of high-resolution urban climate modelling.

**[Summary comment]**

In this manuscript the authors aim to propose a methodological framework for the evaluation of urban climate simulations. The framework is outlined and then tested in high-resolution urban climate modelling simulations over an area encompassing two big cities, Shenzhen and Hong Kong.

The study addresses an important problem that is often overlooked in the urban climate modeling community: the model evaluation. However, the manuscript comes across as a rather superficial and extensive model evaluation of WRF, rather than as a reference for a new model-evaluation framework (see MC1 to MC3). The paper is also poorly written and several sentences are hard to understand (see MC4). The topic of the paper well fits within the scope of GMD, but I would consider it for publication only after major/substantial revisions are performed in line with the MCs below.

**Response:** Thank you for your comments. We have gone through your comments carefully and tried our best to address them one by one. We hope the manuscript has been improved accordingly.

**[Major comment 1]** General MC. This work comes across as a rather descriptive WRF model evaluation rather than as a new model-evaluation framework. The use of PDF and PSS is useful, but I find it is a bit exaggerated to say that a new framework was proposed because these quantities were considered in addition of standard descriptive statistics. This especially in view of the few words spent on the PSS theory in section 2.2 and on the extensive but rather superficial comments made in the model to observation comparison section.

**Response:** Thank you very much for the comment. We didn't find a systematic methodological framework for urban climate model evaluation in previous literature. The PSS theory is a well-known method and really direct and easy to understand, so instead of explaining in details, we pointed the readers to Perkins et al. (2007) for more details.

**[Major comment 2]** Section 2.2: Here I would justify more thoroughly why the authors propose to use a PDF and PSS coefficient when compared to other (perhaps more sophisticated) methods. I would also love to see some physics-based or theoretical derivation for admissible error bounds for given quantities, and a discussion about the strengths and limitations of the proposed framework.

**Response:** Thanks for the comment. This paper is intended to suggest urban climate modelers conduct systematic model evaluations in urban-scale climatology modeling and

provide them a practical methodological framework. We think that the practical and easy-understanding methods are better than complicated and nonintuitive ones. Of course, our methodological framework is not perfect, we have specified the limitations in the discussions section which need to be improved by other scientists in the future.

**[Major comment 3]** I would consider reducing the number of figures and comment more thoroughly.

**Response:** To reduce the number of figures, we restructured Section 4. Moreover, we also moved many redundant figures to Supplementary Material and only kept few example figures in the manuscript. Please refer to the revised manuscript.

**[Major comment 4]** The paper is poorly written and requires substantial revision. Specifically, the authors sometimes use technical terms very loosely (see e.g. mc 8, 9, 10), several sentences are hard to read or understand, and often statements are not supported by proper referencing (see e.g. mc 12, 13). Furthermore, I have encountered several typos and repetitions.

**Response:** Thank you very much for the comment. We did our best to correct these mistakes.

**[Minor comment 1]** P1L27. Consider shortening this sentence.

**Response:** Thank you very much for nice reminder. We revised this sentence as follows:

"Recently, studies on urban climate have received growing attention. It is forecasted that there will be 66% of the world's population living in the urban area by 2050 (United Nations, 2014). The fundamental well-being of the urban population, such as their comfort and health, is directly and significantly affected by urban meteorological conditions, such as temperature, wind speed, and air pollution."[Pg1, Ln25-28]

**[Minor comment 2]** P1L31. Urban climate, and

**Response:** Thanks for your nice reminder. We provided the following citations to support this statement.

Dale, V. H.: The relationship between land-use change and climate change. Ecological applications, 7(3), 753-769, 1997.

Kalnay, E., & Cai, M.: Impact of urbanization and land-use change on climate. Nature, 423(6939), 528, 2003.

**[Minor comment 3]** P2L5. "is more sensitive to the inadequacies of the atmospheric model " -- provide citation to support such a statement.

**Response:** Thanks for your nice reminder. We provided the following citation [Pg2, Ln6] as a support:

Warner, T. T.: Quality assurance in atmospheric modelling. B. Am. Meteorol. Soc., 92(12), 1601-1610, 2011.

**[Minor comment 4]** P2L6. "and the quality of input data" -- provide citation to support such a statement

**Response:** Thanks for your nice reminder. We provided the following citation [Pg2, Ln7] as a support:

Bruyère, C. L., Done, J. M., Holland, G. J., & Fredrick, S.: Bias corrections of global models for regional climate simulations of high-impact weather. Climate Dynamics, 43(7-8), 1847-1856, 2014.

**[Minor comment 5]** P2L10-27. I am glad the authors provided evidence from existing literature.

**Response:** Thanks for the comment.

**[Minor comment 6]** P2L30. "wasn't provided in the previous literatures" -> was not provided.

**Response:** Revised accordingly.

**[Minor comment 7]** P2L30. "It is especially a research gap in 30 urban climate modelling community to proposing a systematic framework and methods for model evaluation." – please rephrase

**Response:** Revised accordingly.

**[Minor comment 8]** P3l32. "Perspectives" -> "periods"?

**Response:** Thank you very much for your nice reminder. We considered what is the appropriate word for this context again. Finally, we changed it to "resolution" [Pg3, Ln30].

**[Minor comment 9]** P3L33. Why direct? Please justify the use of each word, it seems to me the English should be improved

**Response:** Thank you very much for your nice reminder. We revised the sentence as follows:

"In doing so, our instinct can decide whether the modelled results could replicate the temporal and spatial patterns in the observations or not." [Pg3, Ln31-32]

**[Minor comment 10]** P7L29-30. Can you expand and justify why this is the case? Why the adjective "natural"?

**Response:** We already provided an explanation in Subsection 4.1 as follows:

"In fact, the difference includes not only the modelling bias but also an essential difference between a 1-km grid spatial average value and a value of a point located in this grid. Moreover, the observation always locates in an open area, and thus, the observed 2-meters air temperature is the temperature of a point in the open area. The modelled 2-meters air temperature is a mean temperature of a 1-km grid which always includes some vegetation covered areas. It is a common sense that the point air temperature in the open area is always higher than its corresponding 1-km grid mean air temperature in the summertime."

Moreover, we reconsidered if the meaning of the term "natural gap" meets the context of this manuscript. Finally, we replaced it with "essential difference". The essential difference refers to the fact that model outcomes from the simulation models are average values of a grid, while the observations are point-based which only measures the meteorological conditions around the location of the monitoring station.

**[Minor comment 11]** P7L33-34. Related to the previous comment: why is it common sense? Please expand.

**Response:** Thank you. It's a good question. An observation station is always located in an open area – an area without the coverage of trees - and so the measured 2-meter temperature at an observation station is always higher than the modelled temperature since the modelled temperature is calculated as a mean value over a grid with vegetation coverage. We revised the sentence as follows:

"In the summertime, the point air temperature in the open area without coverage of trees is always higher than its corresponding mean air temperature of a 1-km grid with some vegetation coverage." [Pg10, Ln10-12]

**[Minor comment 12]** P10L40. The atmospheric model produces the fine atmospheric features which do not exist in the original meteorological data. – I do not understand what the authors are referring to. Please expand.

**Response:** Thank you very much for the question. Indeed, this sentence is not so easy to understand. We changed it as follows:

"The fine-scale details are constructed by a limited area atmospheric model which consists of physical components driven by the lateral boundary conditions of coarse-scale meteorological data and land surface forcing data (Lo et al., 2008; Hong et al., 2014). However, these details do not exist in the coarse-scale meteorological data (Hong et al., 2014)." [Pg14, Ln17-20]

**[Minor comment 13]** P11L5. What do the authors mean here with "model evaluation"? Please expand.

**Response:** It is a good question. Model evaluation refers to comparisons between the modelled variables and its corresponding observed ones. After modelling, a model evaluation should be conducted for establishing the trustworthiness to the modelling results because of the model incompleteness caused by the approximations and assumptions in the scientific mechanisms of the model even if the model was configured appropriately.

We added these contents into Section 1[Pg1, Ln34-37].

**[Minor comment 14]** P11L41. As it stands to me it does not come across as a sophisticated technique, but a rather as simple approach to evaluate model performance (i.e., look at departures between PDFs between model and observations).

**Response:** Thanks for the comment. This paper was intended to remind (again) urban climate modelers of the necessity of conducting systematic model evaluations in urban-scale climatology modelling and reduce these ambiguous or arbitrary modelling practices. We also provided the urban climate modellers a practical methodological framework for model evaluation which we can not find in the previous literature. We replaced "sophisticated methodological framework" with "practical methodological framework" [Pg14, Ln41] in this sentence.

---

## Author Comment (AC11) · 25 Jul 2019

Manuscript

Please also note the supplement to this comment:
https://www.geosci-model-dev-discuss.net/gmd-2018-220/gmd-2018-220-AC11-supplement.pdf

---

## Author Comment (AC13) · 25 Jul 2019

**Response to Reviewer 4**

**[Cover letter]**

Dear Reviewer,

We appreciate your devoted time in reviewing our paper and your valuable comments which enabled improvements in the current version of the manuscript. The authors have carefully considered all comments and tried our best efforts to address every one of them. However, some revisions may still cannot meet your high standards. The authors welcome further constructive comments if any. Revisions have been made to update the manuscript (highlighted in red) and a detailed point-to-point response is provided below.

Sincerely,
Bo Huang, PhD
bohuang@cuhk.edu.hk
Professor, Department of Geography and Resource Management
The Chinese University of Hong Kong

**[General Comment]** The motivation and objectives of the study are of prime importance, and this revision is a better-organized and streamlined version of the original submission. However, in my opinion, the work still lacks in rigor and depth to be granted publication in GMD (see MC1).

**Response:** Thank you for your comments. It was good news to us that the last revision was improved in your opinion. We have gone through your new comments carefully and tried our best to address them one by one. We hope the manuscript has been improved accordingly. Model evaluation is an essential but overlooked topic. The purpose of this paper is to remind urban climate modelers of the importance of model evaluation and to propose a methodological framework. This framework is not perfect but it is a meaningful beginning. It needs to be developed and supplemented by urban climate modelers of insight in the future.

**[Comment 1]** Section 2.2. The procedure is relatively well described, but the authors do not discuss the criteria for considering a given PSS value acceptable. This I would imagine would vary depending on the quantity of interest, time, and spatial scales of the problem under consideration.

**Response:** Thank you for your reminder. We agree with you that the PSS may change significantly by the quantity of interest, time, and spatial scales in different problems of interest, and so generating a reliable standard of 'acceptable PSS values' cannot be fully dependent on one single study - it has to be a joint effort over time. This study was intended to make a first step in this effort, and the standard will likely improve as more researchers apply the PSS method to many quantities, time, and spatial scales.

*Quantities of Interest*

In light of your comment, we checked the variations of PSS values in our study scope due to different quantities and time of day/year. Figure 1 shows the variations of PSS values due to different quantities of interest. We also checked the statistical significance of the between-group difference using the t-test. P-values among the groups of PSS values for different quantities show that no significant ($p < 0.05$) difference was found among T-2, ST2, RH, and W10, while ST14 and Precip had significantly ($p < 0.05$) lower PSS values compared to the other attributes but the difference between the average levels of the largest and the smallest group was below 0.2. Therefore, it is possible to have a unified standard of acceptable PSS values while highlighting the standard can be relaxed slightly for specific quantities known to have lower reliability.

[Figure]

[Figure]

*Figure 1 Variations in the PSS values due to different quantities (left) and the t-tests among the PSS values for different quantities (right). The red dashed line indicates the 75% quantile level among all PSS values.*

**Time**

We also checked the variations of PSS values over the time of year. PSS values in all months of the year had mean/median PSS values larger than the 75% threshold we proposed. No statistically significant ($p < 0.05$) were observed among any monthly groups of PSS values. It is for future studies to check further how PSS values change over time, for example, ten years ago or later, with significant changes in meteorological contexts.

[Figure]

*Figure 2 Variation of PSS values in different months of the year.*

**Spatial scales**

Since all simulations conducted in this paper use the same spatial extent and scale, we cannot thoroughly check how PSS values vary over different extent and scales. However, it is reasonable to claim that the proposed standard of acceptable PSS values in this study sets the minimum requirement since simulation accuracies were usually found higher for simulations having lower spatial resolutions due to the spatial-smoothing effects. Simulations using coarser spatial resolutions should at least meet our standard of acceptable PSS values, and the standard can be tightened in future studies using coarser spatial resolutions.

**[Comment 2]** English requires substantial revision: Several sentences are qualitative or poorly formulated, and several typos are present throughout.
**Response:** Thank you for the nice reminding. We did our best to correct these errors.

**[Minor comment 1]** P1L30. ". . . urban climate simulation models are among the most powerful ones." –> This sentence is not very accurate. What do the authors mean by "most powerful"?
**Response:** Thanks for your comment. We have changed 'powerful' to 'widely-used.'

**[Minor comment 2]** P1L34. "its corresponding observed ones." –> "and corresponding observations."
**Response:** Thank you very much for the reminder. We have made revisions accordingly.

**[Minor comment 3]** P1L35-37. "Model" or "modeling is used eight times; please rephrase avoiding repetitions.
**Response:** Thank you very much for the reminder. We have made revisions accordingly.

**[Minor comment 4]** P2L1-2. This sentence is a repetition of the concepts explained in the preceding paragraph. I suggest removing it or rephrasing.
**Response:** Thank you very much for the reminder. We rephrased the sentence.

**[Minor comment 5]** P2L10. "of every conclusion" –> "of conclusions".
**Response:** Thank you very much for the reminder. We have made revisions accordingly.

**[Minor comment 6]** P2L35. "interval" –> "departure"?
**Response:** Thank you very much for the reminder. We have made revisions accordingly.

**[Minor comment 7]** P3L30. "instinct" –> A rigorous procedure rather than instinct should be adopted to assess whether model results compare well against experimental measurements.
**Response:** Thanks for your reminder. We have revised the sentence as follows,

"Therefore, we included three different temporal resolutions in our model evaluation framework (Table 1) - annual, monthly, and daily - to provide a sophisticated view on whether the modelled results could replicate the temporal and spatial patterns in the observations or not."

**[Minor comment 8]** Fig 4. "Comparaison" –> "Comparison" (title of the figures)
**Response:** Thank you very much for the reminder. We have made revisions accordingly.

**[Minor comment 9]** P13L28. When comparing point-wise measurements with grid-cell averaged simulation results, some kind of upscaling procedure should also be adopted. Can the author address this problem?

**Response:** Yes, you are right that there exist interpolation methods available to transform point-based observations to grid-based data analytical results. However, doing so will introduce more uncertainty associated with the interpolation method used and the parameter selected or optimized in the interpolation method. Therefore, in this paper, we chose a more explicit path by directly comparing the observations with the grid-based simulations. We also pointed out the potential risks in this comparison for the readers to consider whether to improve their practice adopting methods such as interpolation or not.

**[Minor comment 10]** P13L33. "Theoretically, verifying or validating a model is impossible." –> please specify which "model".

**Response:** The model specified the numerical model in the earth science in the paper of Oreskes et al. 1994 (Oreskes, N., Shrader-Frechette, K., & Belitz, K: Verification, validation, and confirmation of numerical models in the earth sciences, Science, 263(5147), 641-646, DOI: 10.1126/science.263.5147.641, 1994.). The atmospheric model also is a numerical model in earth science. The model specifies the atmospheric model in our manuscript. We added a sentence as follow at the beginning of Subsection 5.1 and revised the "model" to "atmospheric model":

"*The atmospheric model also is one of the earth-scientific numerical models.*"

**Response to Reviewer 5**

**[Cover letter]**

Dear Reviewer,

We appreciate you for spending time to review our paper and providing some valuable comments. It is your valuable and insightful comments that led to possible improvements in the current version. The authors have carefully considered the comments and tried our best efforts to address every one of them. However, some revisions may still cannot meet your high standards. The authors welcome further constructive comments if any. We provided the point-to-point response first and will provide the updated version of the paper after proofreading complete.

Sincerely,

Bo Huang, PhD

bohuang@cuhk.edu.hk

Professor, Department of Geography and Resource Management

The Chinese University of Hong Kong

**[General Comment]**

(1) The authors touch one of the most important problems of model development and using, the model verification workflow. I completely agree with authors that is very important research question, which frequently is often omitted in the model-based research. This scientific problem is even more important for urban climate modelling due to the lack of the urban-scale observations and high complexity of urban climate processes.

**Response:** Thank you for your comments.We are apprectated that you deemed the topic of this study impontant.

(2) However, in my opinion, the manuscript is far away from being accepted. The biggest problem of the manuscript is briefly described in the next few sentences. Starting just from the manuscript's tile, authors refer to the problems of the urban climate research and modelling. However, the presented results poorly fit typical urban climate research framework and do not touch the known problems of urban climate modelling. The urban climatology & meteorology typically works with anomalies such as urban heat island, urban dry/moist islands, urban-induced precipitation anomalies, etc. The state-of-the art

mesoscale models, such as WRF, COSMO or HIRLAM, are still not perfect in terms of accurate simulation of these anomalies. The development of the common evaluation & verification methodology for urban climate is a very relevant research question. However, the presented results deals practically nothing with indicated research problem. The presented results could not tell the reader, how good or bad is the considered model in terms of the simulation of the specific urban climate features. During the previous stages of revision, the authors have provided results focused on urban-rural differences (e.g. Figure 5). However, these figures are still useless for the evaluation of the urban climate modelling, because the observed and modelled values are spaced apart in different subplots.

**Response:** Thank you for your comments. However, we think that the reviewer mixes up two different topics: simulate urban climate features and model evaluation. Urban climate features' simulations are the research topics to seek out new urban climate features by using modelling technologies. The trustworthiness of the results of these research needs to be established by model evaluation. The model evaluation is the comparisons of the modeled meteorological variables with its corresponding observed ones. In this study, we only have five meteorological observed data: air temperature, MODIS surface temperature, 10-m wind speed, 2-m relative humidity, and precipitation, and accordingly we only can conduct the comparison between these observed data with its corresponding modelled ones. The purpose of this paper is to tell the readers the importance of model evaluation which has been overlooked by previous urban climate research and proposed a methodological framework of model evaluation which has been mentioned in previous literature.

(3) The presented results raise many questions even in isolation from the specific problems of urban climate modelling. For example, the authors provide a great amount of the same type of graphs with a seasonal variation of the observed and modelled values. The manuscript and supplementary materials are strongly overloaded by similar graphs. What do they want to show by this plenty of graphs?

**Response:** The model evaluation is the comparisons between modelled variables and its corresponding observed ones. Each modelled variable has a set of graphs. The type of graphs is same, but the variable is different. Therefore, it is necessary to shows all comparisons between modelled variables and its corresponding ones even if the graphics are similar.

(4) It is trivial that regional climate model, forced by the realistic reanalysis data, could reasonably simulate the seasonal cycle of the key weather variables. More interesting and relevant question is how the high-resolution mesoscale model captures the regional climate features.

**Response:** The reviewer mixes up the model evaluation and the research findings in urban cli-matological features. The findings in urban climatological features retrieved from the urban climate modelling results usually cannot evaluate directly. Model evaluation is a critical step of quality assurance to the modelling results. Therefore, model evaluation is a responsibility of climate modellers for establishing the trustworthiness to the findings, which is the reason why we emphasized the importance of the model evaluation in urban climate researches.

(5) The unique dense observational network could provide a lot of information on this topic. However, this part of analysis is omitted in the study. Even the questions related to the diurnal cycle are more relevant due to the well-known biases of the daytime and nighttime biases of the models. But this type of analysis is given much less attention in comparison to the analysis of the seasonal variations.

**Response:** In fact, we provided the comparisons in the daytime and night-time variations be-tween each observed meteorological variable and its corresponding modelled ones.

(6) Finally, it is important to show the advantages of the proposed verification framework and statistical scores. The authors criticize the simpler approaches of the model verification in the introduction. But what benefits do the presented approach give in comparison to the simpler approaches (e.g. the simple biases or model-to-observation plots, which are frequently used in model-based studies)? It would be amazing to present, e.g., that presented framework allows to identify some model errors that could not be revealed by simpler methods. But this is missing. The authors only present the model scores for different variables. But with what these values should be compared?

**Response:** Thank you very much for your comments. It is a good future research direction.

(7) I suggest that the manuscript should be significantly revised before acceptation to GMD. In my opinion, one way of revision is adding more focus to the regional climate features and, specifically, urban climate features such as urban heat island and its quantitative metrics. Otherwise, the authors should not claim about urban climate modelling in the title, abstract and introduction.

**Response:** We don't agree the comments. The model evaluation is different with the researches in regional climate features, such as urban heat island, urban dry/moist islands, urban-induced precipitation anomalies, etc. The model evaluation is a quality assurance to the new findings in regional climate features' researches. The paper intend to remind urban climate researchers the importance of model evaluation and provide a methodological

framework for it.

(8) In addition, the are some other specific issues related, which are listed below. Please, note that these comments are addition to the general comment, but not its detailed explanation. In other words, resolving only the indicated specific questions is not sufficient for the revision.

**Response:** The paper had been revised many times. We are confident that it is valuable to publish in high impact journal before it reveals a pain-point which the urban climate modellers never paid enough attention on the model evaluation.

**[Specific Comment 1]** P1, L30-31: The discussed tools and models are very different in terms of scale and complexity. It will be good to prove some examples of the certain tools and models.

**Response:** It is not the focus of this paper to discuss the difference of tools and models in terms of scale and complexity. This sentence was changed as follows:

*In this vein, many tools have been developed, and the rapidly developing urban climate simulation models are among the most widely-used ones.*

**[Specific Comment 2]** P1, L33 – P2, L8. I completely agree with general idea of the paragraph. However, there is a number of studies, where detailed verification of urban-scale models is performed. This studies should be indicated in the literature review together with studies, where verification part is omitted or not sufficient.

**Response:** We added more details of previous studies which model evaluation part is omitted or not sufficient in Section S10 of Supplementary Material.

**[Specific Comment 3]** P3, L1-21, sect 2.1 (and also P6, L1-9). It is common to present more detailed information about the model setup in such regional modelling studies. E.g. the scheme of the nested domains is missing. For the urban climate modelling studies, it is common to present more detailed information about the study area and land use/land cover data. What are the exact list of the urban land cover parameters, used by the model? What are the typical values of the urban faction and anthropogenic heat flux in the study area? How the used parameters were obtained? These information is required to comparing of the presented study to other urban climate modelling studies. Referring to the dataset name, even without a literature reference, in insufficient.

And more general remark: I suggest to join the two indicated sections to one section, related to urban climate model and its setup.

**Response:** We provided the information of the urban climate model and its setup in Sections S1 and S4 of Supplementary Material. Moreover, we will provided all detail information in other two papers (*A high-resolution urban land surface dataset to investigate the urbanization impact using urban climate modelling: 1979-2010* and *Quality assurance in high-resolution urban climate simulation: using WRF ARW/LSM/SLUCM model (version 3.7.1) as a case study*).

**[Specific Comment 4]** P3, L18-21: Is 1-day spin up sufficient for development of the urban climate features in the model?

**Response:** The 1-day spin up is enough to each 4-days simulation segment.

**[Specific Comment 5]** P8 (Figure 3). Why do you plot modelled values with so low temporal resolution (only 4 values for a day, the same temporal resolution as it is in FNL reanalysis)? The key advantage of the high resolution regional climate model is opportunity to increase the resolution of the driving gridded meteorological data (FNL reanalysis in our case), both in space and in time. Using the model output with 1-hour resolution with be improve the results and the presentation quality.

**Response:** The urban climate simulation is a computational resources consuming job, especially computing wall time and storage space. In this study, one 4-days simulation segment need 1 day node-computing wall time and 28 G storage space. The temporal span of urban climate simulation in this study is one year. Total 122 4-days simulation segment need 122 days node-computing wall time and 3 T G storage space. The simulation job outputs data per hour would be need huge computational resources. Moreover, this study is just for presenting a methodological framework, and accordingly 6-hour temporal resolution is enough.

**[Specific Comment 6]** P8-9 (Figure 4, 5): As I noticed in the general comment, these figures seems to be useless for model evaluation, because the modelled and observed values are displaced to different subplots.

**Response:** We don't agree. For the interpretations of Figures 4 and 5, please refer to Pg6,Ln30 to Pg7, Ln2.

**[Specific Comment 7]** P10, L13-15: Please, clarify, good in comparison to what? E.g. you could compare the presented score by the score, obtained for original gridded data (FNL reanalysis), or by the score obtained by WRF model with different, or with some scores from literature. I

have the same comment to the places in the text where same scores for other variables are presented and discussed.

**Response:** The PSS may change significantly by the quantity of interest, time, and spatial scales in different problems of interest, and so generating a reliable standard of 'acceptable PSS values' cannot be fully dependent on one single study - it has to be a joint effort over time. This study was intended to make a first step in this effort, and the standard will likely improve as more researchers apply the PSS method to many quantities, time, and spatial scales. Moreover, we already explained the reasons why the modelled variable differ from its observed ones in Subsections 5.2.

Other comment to these lines: as I've noticed in general comment, it is trivial that model captures the seasonal cycle, and that the seasonal cycle is more-o-less the urban and rural areas. More attention should be addressed to diurnal cycle and spatial variations within the study area.

**Response:** In fact, we provided the comparisons in the daytime and night-time variations between each observed meteorological variable and its corresponding modelled ones.

**[Specific Comment 8]** P11, L21-23: Please, clarify, acceptable for what? As I've noticed from the Supplementary materials, the mean model bias for the surface temperature could be quite high, up to 10K. Why do you consider it acceptable? The same comment is for the next section related to wind.

**Response:** The PSS may change significantly by the quantity of interest, time, and spatial scales in different problems of interest, and so generating a reliable standard of 'acceptable PSS values' cannot be fully dependent on one single study - it has to be a joint effort over time. This study was intended to make a first step in this effort, and the standard will likely improve as more researchers apply the PSS method to many quantities, time, and spatial scales. Moreover, we already explained the reasons why the modelled variable differ from its observed ones in Subsections 5.2.

---

## Referee Report (RR1)

**Reviewer comments**

**Model evaluation of high-resolution urban climate simulations:**
**using WRF/Noah LSM/SLUCM model (Version 3.7.1) as a case study**

**Recommendation:** Accept with major revisions.

**Summary comment**

In this manuscript the authors aim to propose a methodological framework for the evaluation of urban climate simulations. The framework is outlined and then tested in high-resolution urban climate modelling simulations over an area encompassing two big cities, Shenzhen and Hong Kong. The study addresses an important problem that is often overlooked in the urban climate modeling community: the model evaluation. However, the manuscript comes across as a rather superficial and extensive model evaluation of WRF, rather than as a reference for a new model-evaluation framework (see MC1 to MC3). The paper is also poorly written and several sentences are hard to understand (see MC4). The topic of the paper well fits within the scope of GMD, but I would consider it for publication only after major/substantial revisions are performed in line with the MCs below.

**Major comment**

1. General MC. This work comes across as a rather descriptive WRF model evaluation rather than as a new model-evaluation framework. The use of PDF and PSS is useful, but I find it is a bit exaggerated to say that a new framework was proposed because these quantities were considered in addition of standard descriptive statistics. This especially in view of the few words spent on the PSS theory in section 2.2 and on the extensive but rather superficial comments made in the model to observation comparison section.

2. Section 2.2: Here I would justify more thoroughly why the authors propose to use a PDF and PSS coefficient when compared to other (perhaps more sophisticated) methods. I would also love to see some physics-based or theoretical derivation for admissible error bounds for given quantities, and a discussion about the strengths and limitations of the proposed framework.

3. I would consider reducing the number of figures and comment more thoroughly.

4. The paper is poorly written and requires substantial revision. Specifically, the authors sometimes use technical terms very loosely (see e.g. mc 8, 9, 10), several sentences are hard to read or understand, and often statements are not supported by proper referencing (see e.g. mc 12, 13). Furthermore, I have encountered several typos and repetitions.

**Minor comments**

1. P1L27. Consider shortening this sentence.
2. P1L31. Urban climate,  and
3. P2L5. "is more sensitive to the inadequacies of the atmospheric model " -- provide citation to support such a statement

4. P2L6. "and the quality of input data" -- provide citation to support such a statement
5. P2L10-27. I am glad the authors provided evidence from existing literature.
6. P2L30. "wasn't provided in the previous literatures" -> was not provided.
7. P2L30. "It is especially a research gap in 30 urban climate modelling community to proposing a systematic framework and methods for model evaluation." – please rephrase
8. P3l32. "Perspectives" -> "periods"?
9. P3L33. Why direct? Please justify the use of each word, it seems to me the English should be improved
10. P7L29-30. Can you expand and justify why this is the case? Why the adjective "natural" ?
11. P733-34. Related to the previous comment: why is it common sense? Please expand..
12. P10L40. The atmospheric model produces the fine atmospheric features 40 which do not exist in the original meteorological data. – I do not understand what the authors are referring to. Please expand.
13. P11L5. What do the authors mean here with "model evaluation"? Please expand.
14. P11L41. As it stands to me it does not come across as a sophisticated technique, but a rather as simple approach to evaluate model performance (i.e., look at departures between PDFs between model and observations).

---

## Referee Report (RR2)

**Comments on the revised manuscript "Model evaluation of high-resolution urban climate simulations: using WRF/Noah LSM/SLUCM model (Version 3.7.1) as a case study"**

**General comment:** The authors touch one of the most important problems of model development and using, the model verification workflow. I completely agree with authors that is very important research question, which frequently is often omitted in the model-based research. This scientific problem is even more important for urban climate modelling due to the lack of the urban-scale observations and high complexity of urban climate processes.

However, in my opinion, the manuscript is far away from being accepted. The biggest problem of the manuscript is briefly described in the next few sentences. Starting just from the manuscript's tile, authors refer to the problems of the urban climate research and modelling. However, the presented results poorly fit typical urban climate research framework and do not touch the known problems of urban climate modelling. The urban climatology & meteorology typically works with anomalies such as urban heat island, urban dry/moist islands, urban-induced precipitation anomalies, etc. The state-of-the art mesoscale models, such as WRF, COSMO or HIRLAM, are still not perfect in terms of accurate simulation of these anomalies. The development of the common evaluation & verification methodology for urban climate is a very relevant research question. However, the presented results deals practically nothing with indicated research problem. The presented results could not tell the reader, how good or bad is the considered model in terms of the simulation of the specific urban climate features. During the previous stages of revision, the authors have provided results focused on urban-rural differences (e.g. Figure 5). However, these figures are still useless for the evaluation of the urban climate modelling, because the observed and modelled values are spaced apart in different subplots.

The presented results raise many questions even in isolation from the specific problems of urban climate modelling. For example, the authors provide a great amount of the same type of graphs with a seasonal variation of the observed and modelled values. The manuscript and supplementary materials are strongly overloaded by similar graphs. What do they want to show by this plenty of graphs? It is trivial that regional climate model, forced by the realistic reanalysis data, could reasonably simulate the seasonal cycle of the key weather variables. More interesting and relevant question is how the high-resolution mesoscale model captures the regional climate features. The unique dense observational network could provide a lot of information on this topic. However, this part of analysis is omitted in the study. Even the questions related to the diurnal cycle are more relevant due to the well-known biases of the daytime and nighttime biases of the models. But this type of analysis is given much less attention in comparison to the analysis of the seasonal variations.

Finally, it is important to show the advantages of the proposed verification framework and statistical scores. The authors criticize the simpler approaches of the model verification in the introduction. But what benefits do the presented approach give in comparison to the simpler approaches (e.g. the simple biases or model-to-observation plots, which are frequently used in model-based studies)? It would be amazing to present, e.g., that presented framework allows to identify some model errors that could not be revealed by simpler methods. But this is missing. The authors only present the model scores for different variables. But with what these values should be compared?

I suggest that the manuscript should be significantly revised before acceptation to GMD. In my opinion, one way of revision is adding more focus to the regional climate features and, specifically, urban climate features such as urban heat island and its quantitative metrics. Otherwise, the authors should not claim about *urban* climate modelling in the title, abstract and introduction.

In addition, the are some other specific issues related, which are listed below. Please, note that these comments are addition to the general comment, but not its detailed explanation. In other words, resolving only the indicated specific questions is not sufficient for the revision.

**Other specific comments:**

P1, L30-31: The discussed tools and models are very different in terms of scale and complexity. It will be good to prove some examples of the certain tools and models.

P1, L33 – P2, L8. I completely agree with general idea of the paragraph. However, there is a number of studies, where detailed verification of urban-scale models is performed. This studies should be indicated in the literature review together with studies, where verification part is omitted or not sufficient.

P3, L1-21, sect 2.1 (and also P6, L1-9). It is common to present more detailed information about the model setup in such regional modelling studies. E.g. the scheme of the nested domains is missing. For the urban climate modelling studies, it is common to present more detailed information about the study area and land use/land cover data. What are the exact list of the urban land cover parameters, used by the model? What are the typical values of the urban faction and anthropogenic heat flux in the study area? How the used parameters were obtained? These information is required to comparing of the presented study to other urban climate modelling studies. Referring to the dataset name, even without a literature reference, in insufficient.

And more general remark: I suggest to join the two indicated sections to one section, related to urban climate model and its setup.

P3, L18-21: Is 1-day spin up sufficient for development of the urban climate features in the model?

P8 (Figure 3). Why do you plot modelled values with so low temporal resolution (only 4 values for a day, the same temporal resolution as it is in FNL reanalysis)? The key advantage of the high resolution regional climate model is opportunity to increase the resolution of the driving gridded meteorological data (FNL reanalysis in our case), both in space and in time. Using the model output with 1-hour resolution with be improve the results and the presentation quality.

P8-9 (Figure 4, 5): As I noticed in the general comment, these figures seems to be useless for model evaluation, because the modelled and observed values are displaced to different subplots.

P10, L13-15: Please, clarify, good in comparison to what? E.g. you could compare the presented score by the score, obtained for original gridded data (FNL reanalysis), or by the score obtained by WRF model with different, or with some scores from literature. I have the same comment to the places in the text where same scores for other variables are presented and discussed.

Other comment to these lines: as I've noticed in general comment, it is trivial that model captures the seasonal cycle, and that the seasonal cycle is more-o-less the urban and rural areas. More attention should be addressed to diurnal cycle and spatial variations within the study area.

P11, L21-23: Please, clarify, acceptable for what? As I've noticed from the Supplementary materials, the mean model bias for the surface temperature could be quite high, up to 10K. Why do you consider it acceptable? The same comment is for the next section related to wind.

---

## Referee Report (RR3)

**Reviewer comments for manuscript 2018JD029383**

**Model evaluation of high-resolution urban climate simulations: using WRF/Noah LSM/SLUCM model (Version 3.7.1) as a case study.**

[ **Summary comment** ] The motivation and objectives of the study are of prime importance, and this revision is a better-organized and streamlined version of the original submission. However, in my opinion, the work still lacks in rigor and depth to be granted publication in GMD (see MC1).

[**Comment 1**] Section 2.2. The procedure is relatively well described, but the authors do not discuss the criteria for considering a given PSS value acceptable. This I would imagine would vary depending on the quantity of interest, time and spatial scales of the problem under consideration.

[**Comment 2**] English requires substantial revision: Several sentences are qualitative or poorly formulated, and several typos are present throughout.

[**Minor comment 1**] P1L30. "...urban climate simulation models are among the most powerful ones." –> This sentence is not very accurate. What do the authors mean by "most powerful"?

[**Minor comment 2**] P1L34. "its corresponding observed ones." –> "and corresponding observations."

[**Minor comment 3**] P1L35-37. "Model" or "modeling is used 8 times, please rephrase avoiding repetitions.

[**Minor comment 4**] P2L1-2. This sentence is a repetition of the concepts explained in the preceding paragraph. I suggest removing it or rephrasing.

[**Minor comment 5**] P2L10. "of every conclusion" –> "of conclusions".

[**Minor comment 6**] P2L35. "interval" –> "departure"?

[**Minor comment 7**] P3L30. "instinct" –> A rigorous procedure rather than instinct should be adopted to assess whether model results compare well against experimental measurements.

[**Minor comment 8**] Fig 4. "Comparaison" –> "Comparison" (title of the figures)

[**Minor comment 9**] P13L28. When comparing point-wise measurements with grid-cell averaged simulation results, some kind of upscaling procedure should also be adopted. Can the author address this problem?

[**Minor comment 10**] P13L33. "Theoretically, verifying or validating a model is impossible" –> please specify which "model".